# Active species in chloroaluminate ionic liquids catalyzing low-temperature polyolefin deconstruction

Wei Zhang [1,2] ✉, Rachit Khare[2], Sungmin Kim [1], Lillian Hale[1], Wenda Hu[1], Chunlin Yuan[2], Yaoci Sheng[2], Peiran Zhang [2], Lennart Wahl[2], Jiande Mai[1], Boda Yang[1], Oliver Y. Gutiérrez [1], Debmalya Ray[1], John Fulton[1], Donald M. Camaioni[1], Jianzhi Hu [1,3], Huamin Wang[1], Mal-Soon Lee [1] & Johannes A. Lercher [1,2] ✉

Chloroaluminate ionic liquids selectively transform (waste) polyolefins into gasoline-range alkanes through tandem cracking-alkylation at temperatures below 100 °C. Further improvement of this process necessitates a deep understanding of the nature of the catalytically active species and the correlated performance in the catalyzing critical reactions for the tandem polyolefin deconstruction with isoalkanes at low temperatures. Here, we address this requirement by determining the nuclearity of the chloroaluminate ions and their interactions with reaction intermediates, combining in situ $^{27}Al$ magic-angle spinning nuclear magnetic resonance spectroscopy, in situ Raman spectroscopy, Al K-edge X-ray absorption near edge structure spectroscopy, and catalytic activity measurement. Cracking and alkylation are facilitated by carbenium ions initiated by $AlCl_3$-tert-butyl chloride (TBC) adducts, which are formed by the dissociation of $Al_2Cl_7^-$ in the presence of TBC. The carbenium ions activate the alkane polymer strands and advance the alkylation cycle through multiple hydride transfer reactions. In situ $^1H$ NMR and operando infrared spectroscopy demonstrate that the cracking and alkylation processes occur synchronously; alkenes formed during cracking are rapidly incorporated into the carbenium ion-mediated alkylation cycle. The conclusions are further supported by ab initio molecular dynamics simulations coupled with an enhanced sampling method, and model experiments using n-hexadecane as a feed.

Upcycling discarded polyolefins into a range of products that align with existing end markets is a critical step towards achieving a circular economy[1,2]. Polyolefin waste could be a promising feedstock for a next-generation refinery, with volumes ( ~160 Mt/yr⁻¹; 2019) equivalent to 1 billion barrels of crude oil refining capacity[3,4].

Current catalytic technologies to convert spent polyolefins into transportation fuels rely on catalytic pyrolysis and hydrocracking over solid acid catalysts, for which significant energy input at high temperatures (T > 300 °C) is required to overcome the endothermic C-C cleavage[5–7]. To overcome thermodynamic limitations, recent

[1]Institute for Integrated Catalysis, Pacific Northwest National Laboratory, P.O. Box 999, Richland WA, USA. [2]Department of Chemistry and Catalysis Research Center, Technische Universität München, Lichtenbergstr. 4, Garching, Germany. [3]The Gene and Linda Voiland School of Chemical Engineering and Bioengineering, Washington State University, Pullman, WA, USA. ✉e-mail: wzhangx@outlook.com; johannes.lercher@ch.tum.de

**Fig. 1 | Tandem cracking-alkylation for converting polyolefins to liquid alkanes with chloroaluminate ionic liquids.** Schematic illustration of the endothermic C-C bond cleavage of polyolefins, kinetically coupled with the exothermic alkylation of the resulting olefin intermediate, enabling low-temperature conversion of polyolefins into liquid alkanes in ionic environments.

approaches utilize hydrocracking or hydrogenolysis[8–13], tandem hydrogenolysis-aromatization[14], dehydrogenation-metathesis[15–18]. These methods achieve conversions beyond the equilibrium conversion for cracking at low temperatures. However, these tandem processes still require moderate to high reaction temperatures (~200–250 °C) and suffer from severe catalyst deactivation by coking[19,20].

Recently, we reported a low-temperature strategy for upgrading discarded polyolefins to gasoline-range iso-alkanes in a single-stage process catalyzed by an acidic chloroaluminate-based ionic liquid (Fig. 1)[21]. This approach kinetically couples the endothermic cleavage of the polymer C−C bonds with the exothermic alkylation of iso-paraffins, hence, thermodynamically allowing full conversion of the polymer below 100 °C. However, the identification of the active catalyst speciation in this complex transformation has remained unresolved. It should be noted that this represents a longstanding question in the study of chloroaluminate-based ionic liquids for reaction such as paraffin alkylation, despite the extensive research and the well-established industrial application[22,23].

Chloroaluminate ionic liquids mainly consist of $AlCl_4^-$ and $Al_2Cl_7^-$ anions. The mononuclear $AlCl_4^-$ is chemically relatively inert[24]. Consequently, its adduct with $AlCl_3$, i.e., $Al_2Cl_7^-$ has been suggested to be the catalytically active site[25]. However, our preceding kinetic analysis showed that the initial low-density polyethylene (LDPE) conversion rate was not proportional to the concentration of $Al_2Cl_7^-$. We then hypothesize that a small concentration of $AlCl_3$ from the dissociation of $Al_2Cl_7^-$ (Supplementary Fig. 1) functions as an ephemeral transition structure[21]. This monomeric $AlCl_3$ can not be detected analytically, but its concentration can be derived via the equilibrium constants as it is in equilibrium with $AlCl_4^-$ and $Al_2Cl_7^-$. Additionally, alkyl chloride additives are essential in chloroaluminate-based ionic liquid-catalyzed alkylation of isoparaffins with olefins. Numerous studies have

attributed the effectiveness of these additives to the in-situ generation of HCl[26], functioning similarly to classical Brønsted acid-catalyzed alkylation processes, such as those involving hydrofluoric acid (HF) or sulfuric acid ($H_2SO_4$)[27]. Jess and coworker showed that halide additives are the carbenium ion initiators that directly generate the carbenium ions for the alkylation reaction, in contrast to Brønsted-acid, where this species is indirectly formed through a hydride shift between protonated butene and isobutane[28]. The challenge lies in elucidating the dual roles of chloroaluminate ions, especially speculative monomeric $AlCl_3$, and the alkyl chloride additive, in the carbenium ion-based mechanism that drives the interconnected cracking and alkylation cycles.

In this study, we address questions concerning the speciation of chloroaluminate species and their role in the low-temperature activation and subsequent tandem cracking-alkylation for polyolefin conversion in the presence of iso-paraffins. We elucidate the nature and structural dynamics of chloroaluminate species within ionic liquids, employing a combination of ${}^{27}Al$ magic-angle spinning nuclear magnetic resonance spectroscopy (${}^{27}Al$ MAS NMR), in situ Raman spectroscopy, and Al K-edge X-ray absorption near edge structure spectroscopy. It is shown that the reaction is started by small amounts of TBC reacting with in situ generated $AlCl_3$ by dissociation of $Al_2Cl_7^-$, providing the initial carbenium ions for the chain process. It should be emphasized that the carbenium ions remain always associated with the formed $AlCl_4^-$ anions. The initial *tert*-butyl carbenium ions activate the C−H bonds of alkanes (LDPE and iC$_5$) via hydride transfer. Kinetic control experiments, along with the atomistic simulations, using linear hexadecane (n−C$_{16}$) as probe molecules are used to elucidate the reaction mechanisms.

## Results
### Synthesis and characterization of ionic liquids
We first identify the structure of the chloroaluminate species and establish qualitative methods to monitor them under operation conditions. Chloroaluminate ionic liquids were prepared by mixing anhydrous $AlCl_3$ with N-butyl pyridinium chloride at various ratios. The samples are denoted as [C$_4$Py]Cl−xAlCl$_3$, wherein x represents the molar ratio of $AlCl_3$ and [C$_4$Py]Cl (see the experimental section for details). The structure of chloroaluminate species is determined by Al K-edge X-ray absorption near edge structure (XANES) and k²-weighted Fourier-transformed extended X-ray absorption fine structure (FT-EXAFS) of [C$_4$Py]Cl-xAlCl$_3$ ionic liquids with varying [C$_4$Py]Cl:AlCl$_3$ molar ratios (Supplementary Fig. 2). The chloroaluminate speciation of [C$_4$Py]Cl-xAlCl$_3$ comprises monomeric $AlCl_4^-$ and dimeric $Al_2Cl_7^-$ anions. In the monomeric $AlCl_4^-$ species, the Al atom is symmetrically coordinated to four chloride ions with identical Al−Cl bond lengths (~2.16 Å). In the dimeric $Al_2Cl_7^-$, the coordination structure around Al is asymmetric. The Al−Cl bond length for the six terminal chloride ions is ~2.12 Å, whereas for the bridging chloride ion (Cl$_{bridge}$), it is ~2.29 Å, indicating distorted tetrahedral symmetry. More details can be found in supplementary note 1, including the systematic increase in the Debye-Waller factor through EXAFS fitting analysis (see Supplementary Figs. 3–6 and Supplementary Table 1).

The chloroaluminate species in the ionic liquids were further identified using ${}^{27}Al$ MAS NMR spectroscopy (Fig. 2a). The spectrum for x = 1 ([C4Py]Cl:1AlCl$_3$) only showed a sharp signal at ~103 ppm, which is assigned to monomeric $AlCl_4^-$[29] Increasing the fraction of anhydrous $AlCl_3$ (x > 1) broadens the resonance signal, accompanied by a new peak at ca. 97 ppm. This peak is attributed to the formation of dimeric $Al_2Cl_7^-$ in equilibrium with $AlCl_4^-$[30]. The increase of the mole ratio to 2.5 ([C4Py]Cl:2.5AlCl$_3$) led to a signal for solid aluminum chloride (~0 ppm). We hereby focus on the homogenous [C$_4$Py]Cl-AlCl$_3$ ionic liquids, with a mole ratio below 2.5, for catalysis in which monomeric $AlCl_4^-$ and dimeric $Al_2Cl_7^-$ dominate.

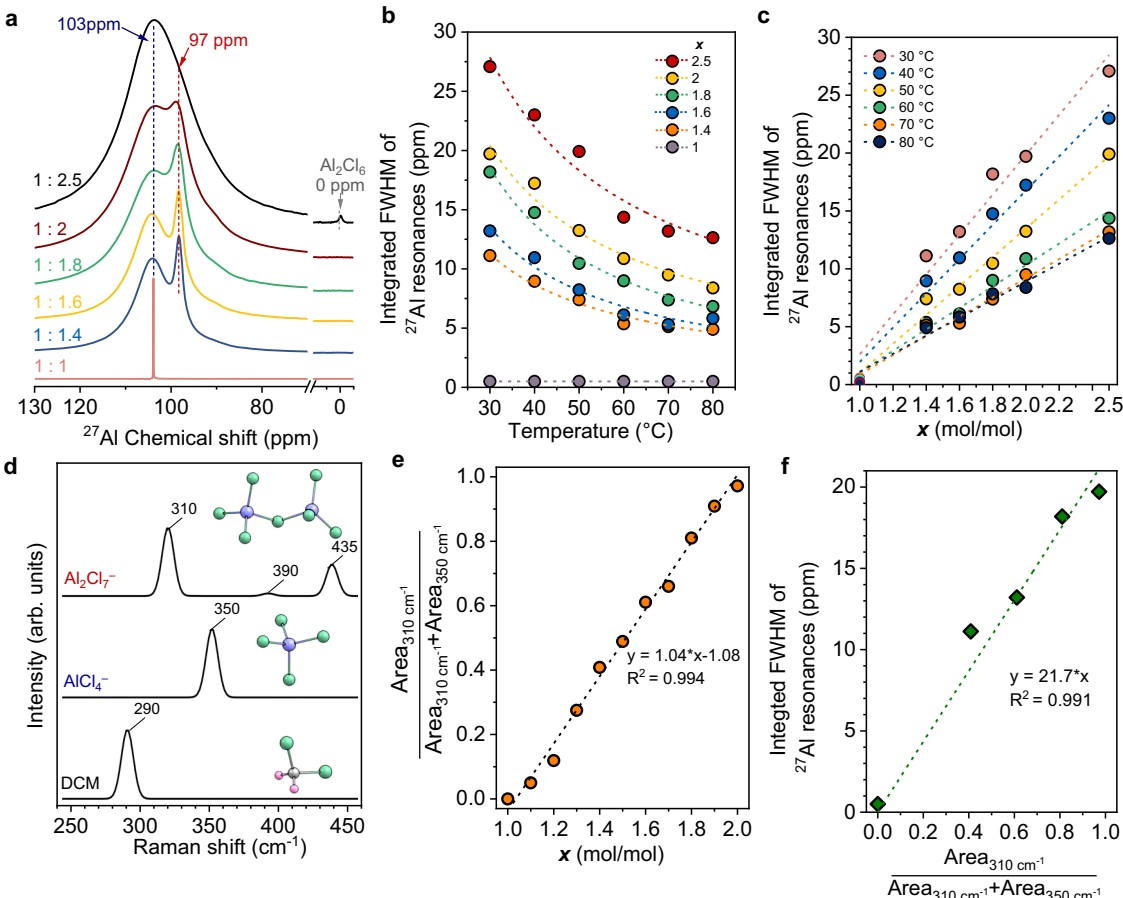

**Fig. 2 | 27Al MAS NMR and Raman spectroscopy on chloroaluminate ionic liquids. a** $^{27}$Al MAS NMR spectra of [C$_4$Py]Cl·$x$AlCl$_3$ ($x$ = 1-2.5). **b** Integrated FWHM of $^{27}$Al resonances as a function of temperatures over [C$_4$Py]Cl·$x$AlCl$_3$. **c** Integrated FWHM of $^{27}$Al resonances as a function of mole ratio of AlCl$_3$/[C$_4$Py]Cl ($x$) at different temperatures. **d** DFT-simulated Raman spectra of AlCl$_4^-$, Al$_2$Cl$_7^-$, and DCM. **e** Plot of the Raman area ratio of Al$_2$Cl$_7^-$ versus the molar ratio of AlCl$_3$/ [C$_4$Py]Cl ($x$); (Note that the area ratio of Raman signals at 310 cm$^{-1}$ and 350 cm$^{-1}$ depends closely on the molar ratio of Al$_2$Cl$_7^-$ and AlC$_4^-$, respectively, derived from the Raman spectra in Supplementary Fig. 9). **f** Relationship between integrated FWHM of $^{27}$Al resonances and the Raman area ratio of Al$_2$Cl$_7^-$ at 30 °C.

We further performed $^{27}$Al NMR at varying temperatures from 30 °C to 80 °C for identify the chloroaluminate species. As shown in Supplementary Fig. 7a, the sharp signal of [C$_4$Py]Cl-1AlCl$_3$ was consistent at different temperatures, indicating that the mononuclear AlCl$_4^-$ species are chemically inert. In the case of ionic liquids with an excess of anhydrous AlCl$_3$ to [C$_4$Py]Cl ($x$ > 1, Supplementary Fig. 7, b-f), the two peaks at 103 and 97 ppm broadened with increasing aluminum content without a significant chemical shift. Numerous studies directly assigned the resonances at 103 and 97 ppm to AlCl$_4^-$ and Al$_2$Cl$_7^-$, respectively[31-34]. However, the two individual signals cannot be directly related to the concentrations of Al$_2$Cl$_7^-$ and AlCl$_4^-$ [35]. A dynamic interconversion between Al$_2$Cl$_7^-$ and AlCl$_4^-$ broadens linewidths, contributing to the resonance assigned to AlCl$_4^-$ at around 103 ppm[36]. This is supported by DFT−NMR calculations (Supplementary Fig. 8a and b). The integrated line widths quantified by the full width at half maximum (FWHM) as a function of temperatures directly reflected this trend (Supplementary Fig. 8c)[37]. Fig. 2b shows that the two resonances at 103 and 97 ppm gradually narrowed with increasing temperature. The integrated FWHM of $^{27}$Al resonances increased linearly with the molar ratio of AlCl$_3$/[C$_4$Py]Cl with good correlations (R$^2$ > 0.98) among all spectra with varying temperatures (Fig. 2c). We infer that increasing temperature intensified the rate of molecular motions and the interconversion between Al$_2$Cl$_7^-$ and AlCl$_4^-$, thereby exhibiting better-defined resonance signals[38,39].

The Raman spectra of the [C$_4$Py]Cl·$x$AlCl$_3$ ionic liquids (Fig. 2d and Supplementary Fig. 9) show a characteristic peak at 350 cm$^{-1}$, which is attributed to the symmetric Cl−Al−Cl stretch of AlCl$_4^-$ anions[40]. The intensity of the AlCl$_4^-$ peak decreased gradually with increasing concentrations of anhydrous AlCl$_3$, accompanied by the increase of the peaks at 310 cm$^{-1}$ and 430 cm$^{-1}$ that are assigned to the symmetric Al−Cl−Al and terminal Al−Cl stretching of dimeric Al$_2$Cl$_7^-$ anions, respectively[41]. The observation suggests that anhydrous AlCl$_3$ drives the transformation from AlCl$_4^-$ and Al$_2$Cl$_7^-$. The ratio of the areas of the signals at 310 cm$^{-1}$ and 350 cm$^{-1}$ (Fig. 2e) mirrors the molar ratio of Al$_2$Cl$_7^-$ to AlCl$_4^-$ [41-43]. The FWHM of $^{27}$Al resonances against area ratios (Fig. 2f) showed a linear correlation (R$^2$ = 0.991).

The combination of in situ $^{27}$Al MAS NMR and Raman spectroscopy establishes, therefore, a quantitative relation of the formation of monomer and dimer aluminate species in ionic liquids that enables also monitoring the species under operation conditions. The changes in concentrations of AlCl$_4^-$ and Al$_2$Cl$_7^-$ (Al$_2$Cl$_7^-$→ AlCl$_4^-$ + AlCl$_3$) over various time intervals (t) simplified as $\sum[AlCl_3]_t = \Delta[AlCl_4^-]_t$, and the Al mole fraction ($\chi_{Al}$) of chloroaluminate species can be expressed as Equations (Eqs. 1–2) (see detailed derivations in Supplementary Note 2):

$$\chi_{Al}\left(\sum[AlCl_3]\right)_t = \chi_{Al}\left(AlCl_4^-\right)_t = \frac{1}{2\left(\frac{Area_{310cm^{-1}}}{Area_{350cm^{-1}}}\right)_t + 2} \quad (1)$$

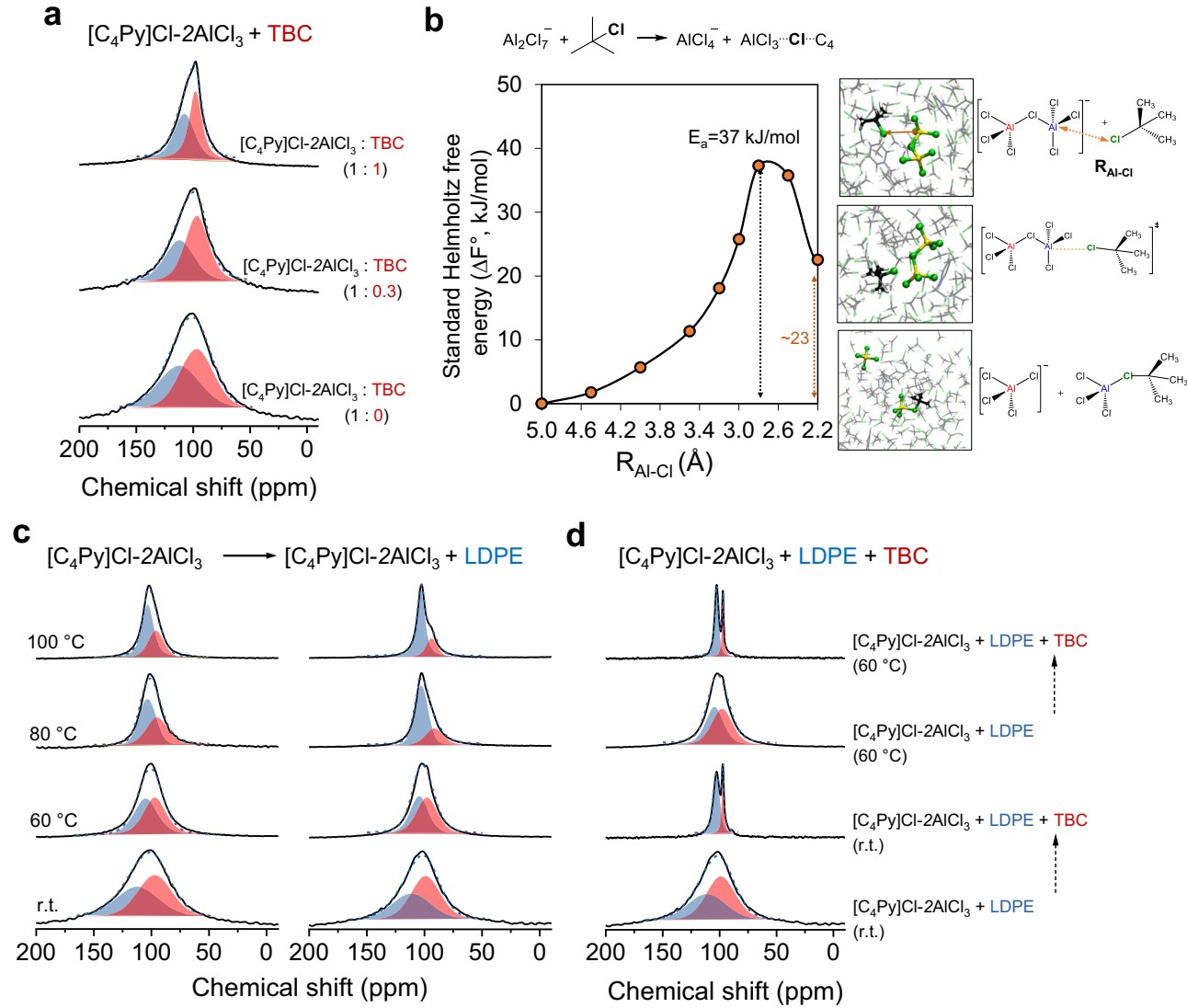

**Fig. 3 | In situ $^{27}$Al MAS NMR on [C4Py]Cl-2AlCl3 catalyzed LDPE deconstruction. a** Recorded of [C$_4$Py]Cl-2AlCl$_3$ reacting with TBC with varying [C$_4$Py]Cl-2AlCl$_3$/TBC ratio from 0 to 1. **b** The computational simulations of Al$_2$Cl$_7^-$ reacting with TBC to form AlCl$_3$-TBC adduct. The standard Helmholtz free energies (ΔF°), dependent on the Al–Cl distance (R$_{Al–Cl}$) between two molecules, are accompanied by atomic configurations that depict the initial, transition, and intermediate states. These data were derived through ab initio molecular dynamics simulations in conjunction with the Blue Moon ensemble method. **c** Recorded during temperature-programmed reaction of LDPE over [C$_4$Py]Cl-2AlCl$_3$ from room temperature (r.t.) to 100 °C; **d** Recorded by adding TBC to the [C$_4$Py]Cl-2AlCl$_3$ catalyzed LDPE reaction at room temperature and 60 °C, respectively.

$$\chi_{Al}\left(Al_2Cl_7^-\right)_t = 1 - \frac{1}{\left(\frac{Area_{310cm^{-1}}}{Area_{350cm^{-1}}}\right)_t + 1} \qquad (2)$$

### Variation of the chloroaluminate species during the reaction

In situ $^{27}$Al NMR experiments were then used to monitor the restructuring of the chloroaluminate species during the reaction. Previous experiments showed that the significant enhancement in LDPE conversion observed with TBC addition[21]. We now probe the interaction of the chloroaluminate species with TBC, the carbenium ion initiator[28], and reaction intermediates during LDPE conversion. Figure 3a shows that the addition of TBC decreased the linewidth of Al signals. This indicates that Al$_2$Cl$_7^-$ and TBC react, forming a more symmetric species by chloride abstraction. Although the linewidths of Al decreased upon the addition of a molar aliquot of TBC compared to when it constituted only one-third, the reduction was not significant. This suggests that the equilibrium is shifted toward the more symmetric species as the relative concentration of TBC increases.

Utilizing ab initio molecular dynamics (AIMD) simulations coupled with a Blue Moon ensemble method (Fig. 3b), we found that the reaction between Al$_2$Cl$_7^-$ and TBC to yield AlCl$_4^-$ and the AlCl$_3$-TBC adduct (Al$_2$Cl$_7^-$ + TBC ⇌ AlCl$_4^-$ + AlCl$_3$···TBC) had a significantly lower energy barrier of 37 kJ/mol compared to direct Al$_2$Cl$_7^-$ dissociation (Al$_2$Cl$_7^-$ ⇌ AlCl$_4^-$ + AlCl$_3$), which exhibits a higher activation energy of 59 kJ/mol (Supplementary Fig. 10). The absence of detected adducts is attributed to interference from the multinuclearity of chloroaluminate anions in chloroaluminate ionic liquids, as the formation of monomeric AlCl$_3$ adducts, typically detectable at ~99 ppm[44,45].

Figure 3c shows the temperature-programmed in situ $^{27}$Al MAS NMR of [C$_4$Py]Cl-2AlCl$_3$ in the presence of LDPE, compared with pure ionic liquids. Upon adding LDPE, the Al resonances at 103 and 97 ppm were well preserved and slightly narrowed with increasing temperatures. This observation suggests that the direct deconstruction of LDPE is limited due to endothermic C-C cleavage. Further adding TBC

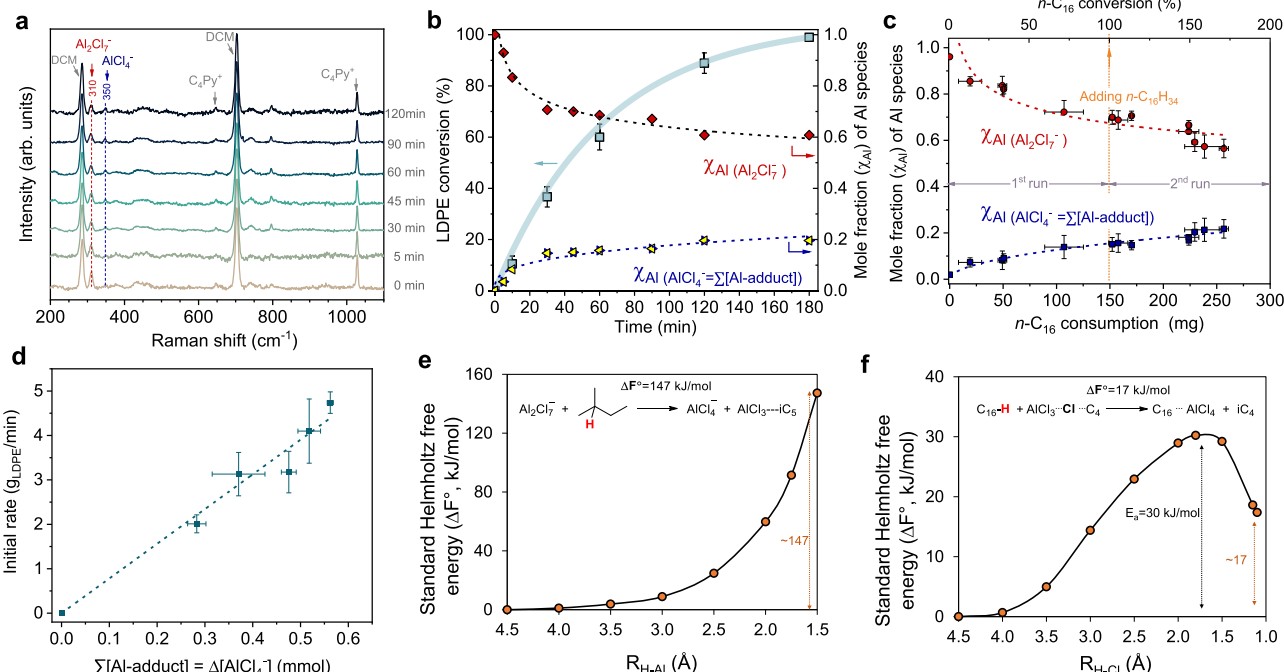

**Fig. 4 | In situ Raman spectra on [C4Py]Cl-2AlCl3 reacting with LDPE and n-hexadecane as a model reactant. a** In situ Raman spectra recorded during the cracking-alkylation of LDPE and iC$_5$ in the presence of TBC additive. **b** The time-resolved conversion profile of LDPE and the corresponding variation of chloroaluminate species using Equation (Eq. (1)). **c** The n-C$_{16}$ consumption as a function of chloroaluminate species in consecutive processes. **d** the initial rate (calculated as grams of LDPE converted per minute) as a function of Al-adducts concentration. Note: we studied the rates at low conversions <20 wt. %. The changes in concentrations over various time intervals are expressed as ∑[Al-adduct] = Δ[AlCl$_4^-$]. Conditions: a-b: LDPE, 200 mg; iC$_5$, 800 mg; [C4Py]Cl·2AlCl3, 2 mmol; TBC, 5 mg; DCM, 3 ml; and temperature, 70 °C. c: In the 1$^{st}$ run, 150 mg of n-C$_{16}$, 600 mg of iC$_5$ and 5 mg of TBC as an additive were added into the reactor containing 2 mmol of [C4Py]Cl·2AlCl3 and 3 ml of DCM. Upon n-C$_{16}$ was fully converted, the reactor was rapidly cooled under −30 °C and replenish with another 150 mg of fresh n-C$_{16}$. **e-f** The standard Helmholtz free energy (ΔF°) of Al$_2$Cl$_7^-$ catalyzed hydride abstraction from iC$_5$ (**e**) and AlCl$_3$-TBC adduct catalyzed hydride abstraction from n-C$_{16}$ (**f**) as a function of the molecular distance between reacting molecules, achieving via ab initio molecular dynamics simulations coupled with a Blue Moon ensemble method.

as a co-reagent with LDPE significantly narrowed the Al resonances, which points to a significant depletion of Al$_2$Cl$_7^-$ species (Fig. 3d). This observation is further supported by a series of spectra obtained at 60 °C.

Therefore, we deduced that the reaction is initiated by TBC reacting with AlCl$_3$ that is in situ generated from the dissociation of Al$_2$Cl$_7^-$, forming the AlCl$_3$-TBC adduct (AlCl$_3$···Cl···iC$_4$). The adducts readily undergo a transformation into reactive ion-pair species (i.e., carbenium ions and AlCl$_4^-$), activating the reacting molecules via consecutive hydride transfer and facilitating the propagation of carbenium ions. Notably, the formed carbenium ions are not bare cations but are surrounded and stabilized by the counterpart AlCl$_4^-$ anions, existing in the form of stable ion pairs[46]. This can explain why the initial reaction rate is directly correlated with the concentration of TBC loading and dissociation of Al$_2$Cl$_7^-$(Supplementary Fig. 11).

The dynamic transformation of these chloroaluminate species within ionic liquids can also be monitored through in situ Raman spectroscopy throughout the reaction. The Raman spectra of [C4Py]Cl-xAlCl3 in the presence and absence of TBC were nearly identical (Supplementary Fig. 12). This further implies that the resulting Al species predominantly exist as organic adducts. Otherwise, the presence of AlCl$_4^-$ (along with carbenium ions) would be reflected by a distinct Raman signal specific to AlCl$_4^-$.

Then, we conducted time-resolved in situ Raman spectroscopy of [C4Py]Cl-2AlCl3 in the presence of LDPE and iC$_5$ together with TBC additive at 70 °C (Fig. 4a). The characteristic band of Al$_2$Cl$_7^-$ at 310 cm$^{-1}$ gradually weakens with reaction time, accompanied by an increase in the new characteristic band of AlCl$_4^-$ (350 cm$^{-1}$). These variations in Al$_2$Cl$_7^-$ and AlCl$_4^-$ concentration are used to quantitatively evaluate the

concentration of formed AlCl$_3$ (which exists as an adduct). Clearly, the variation of χ(Al) of chloroaluminate species showed a close correlation with LDPE conversion (Fig. 4b). Initially, the Al$_2$Cl$_7^-$ species was rapidly consumed, and its fraction decreased by ~0.3 when 40% of LDPE was converted. The remaining LDPE (60%) only consumed ~0.1 of Al$_2$Cl$_7^-$ (Supplementary Fig. 13). This was also observed with the variation of Al$_2$Cl$_7^-$ and AlCl$_4^-$ using n-C$_{16}$ as a reactant instead of LDPE (Supplementary Fig. 14). Particularly in consecutive runs, the chloroaluminate components were kept constant during the conversion of n-C$_{16}$ (Fig. 4c). This stabilization is attributed to the generation of catalytically active species originating from the dissociation of Al$_2$Cl$_7^-$ during the initial stages of the reaction, which then rapidly reaches equilibrium. This process enables the propagation of carbenium ions through consecutive hydride transfers between LDPE and iC$_5$. We found that the initial reaction rate is proportional to the concentration of ∑[Al-adduct] (Fig. 4d), suggesting AlCl$_3$-complex formed in situ through the dissociation of Al$_2$Cl$_7^-$ functions as the catalyst.

The AIMD simulations revealed that the hydride transfer between Al$_2$Cl$_7^-$ and hydrocarbons (e.g., iC$_5$) is a strongly endothermic process, continuously increasing up to 147 kJ/mol (Fig. 4e and Supplementary Fig. 15). This suggests that the Al$_2$Cl$_7^-$ species is not a catalytically active species and, therefore, is hypothesized not to catalyze the hydride transfer. In contrast, the hydride transfer from n-C$_{16}$ to the AlCl$_3$-TBC complex is only 17 kJ/mol (Fig. 4f and Supplementary Fig. 16). The iC$_4^+$ formed from TBC can easily attract/accept a hydride of the PE chain, resulting in the formation of iC$_4$ and PE$^+$-AlCl$_4^-$ adducts (AlCl$_3$···Cl···PE). In the transition state (TS), the proximity of PE to the AlCl$_3$-TBC complex leads to the interaction between PE and iC$_4^+$ and the formation of AlCl$_4^-$ with an activation energy of 30 kJ/mol. These simulation

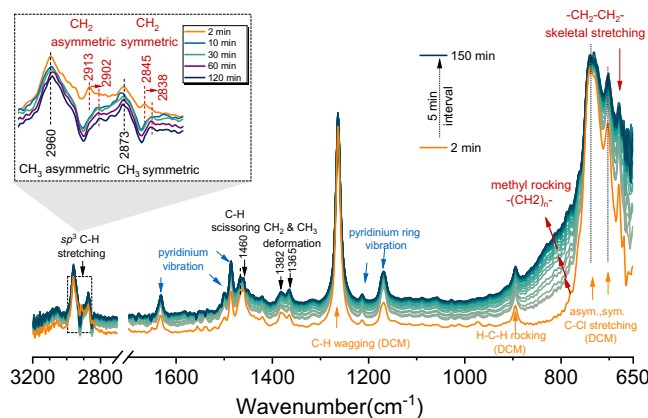

**Fig. 5 | Following the deconstruction of LDPE Operando IR study of conversion over ionic liquids.** Operando IR spectra recorded in the presence of iC$_5$ over [C$_4$Py] Cl-2AlCl$_3$. Conditions: LDPE, 2 g; iC$_5$, 8 g; DCM, 30 ml; [C$_4$Py]Cl-2AlCl$_3$, 20 mmol; TBC 0.5 mmol, and temperature, 60 °C. Data were acquired with an automatic sample scan interval of 15 s over the range of 4000-700 cm$^{-1}$.

results agree well with the in situ NMR and Raman spectroscopy, which indicate that the AlCl$_3$-TBC adduct (AlCl$_3$···Cl···iC$_4$) acts as an active species. It induces intramolecular hydride transfer at the initial stage, providing the initial carbenium ions necessary to start the reaction.

**Mechanism and kinetics of catalytic deconstruction of LDPE via tandem cracking-alkylation**

It is important to note that alkenes are hardly observed in the product stream; only trace amounts of propene (less than 0.1 wt. %) were observed in the headspace during the cracking-alkylation of LDPE and iC$_5$. We inferred that cracking and alkylation are kinetically coupled and proceed, thus, with identical rates. The activated hydrocarbon strands of polymers will crack at positions determined by the rates of hydride shift within the chain. Increasing the hydride shift rate compared to cracking will, thus, lead to longer cracked fragments and heavier products; on the contrary, the relatively higher rate of cracking will give more alkenes, facilitating the formation of acid-soluble oil via oligomerization (as the alkylation is a significantly slower reaction than the oligomerization of the alkenes).

To monitor alkene formation, we employed in situ $^1$H NMR spectroscopy for tandem-cracking-alkylation, using n-C$_{16}$ as a model reactant with iC$_5$ in the presence of TBC additive over [C$_4$Py]Cl-2AlCl$_3$ (Supplementary Fig. 17). We found that the variation of unsaturated hydrogen atoms within the olefinic region (4.4–6.5 ppm) indicates the presence and fluctuation of olefinic intermediates during the reaction. The ratio of unsaturated chains remains constant, maintaining a low concentration of -0.2–0.5% throughout the reaction. To further explore the impact of olefins on the tandem cracking-alkylation of LDPE and iC$_5$, isopentene (C$_5^=$) was introduced instead of the TBC additive (Supplementary Fig. 18). However, this resulted in only 18 wt.% conversion to LDPE, showing no noticeable improvement compared to baseline conditions lacking such additives (Supplementary Fig. 19). Our AIMD-based Blue Moon ensemble simulation confirmed this by showing a highly endothermic process (ΔF° = 125 kJ/mol) responsible for forming the AlCl$_4^-$ and AlCl$_3$-C$_5^=$ adduct through interaction between Al$_2$Cl$_7^-$ and C$_5^=$ (see Supplementary Fig. 20).

Obviously, C$_5^=$ exhibits a pronounced preference for oligomerization over alkylation, which is in agreement with chloroaluminate ionic liquid-catalyzed olefin oligomerization[28]. Thus, we conclude that alkenes, generated synchronously during the cracking cycle, engage in very rapid alkylation reactions with iC$_5$. This interaction significantly alters the equilibrium via the exothermic alkylation of alkene fragments, thus, effectively suppressing olefin oligomerization.

For real-time monitoring of these reactions, we then employed operando infrared spectroscopy, which provided insights into the transformations occurring within the LDPE substrate during the reaction (Supplementary Fig. 21). The time-resolved IR spectra of LDPE in the presence of ionic liquids at 60 °C displayed characteristic LDPE bands (Supplementary Fig. 22). The sp$^3$ C–H vibrations at 2913 cm$^{-1}$ and 2845 cm$^{-1}$, attributed to $\nu_{as}$(CH$_2$) and $\nu_s$(CH$_2$) of LDPE chains[47], respectively, increased in intensity with time (Supplementary Fig. 22 a and b). This increase indicates that more of the polyolefin was observable at the monitoring window of the reaction cell, i.e., the spectroscopically accessible volume increased. Notably, the appearance of weak signals corresponding to terminal CH$_3$ groups indicates the limited LDPE deconstruction, attributable to the endothermic cleavage of the C–C bonds, which is thermodynamically unfavorable at such low temperatures. This observation is corroborated by in situ $^{27}$Al NMR spectra recorded for LDPE deconstruction (Supplementary Fig. 22c), reinforcing the conclusion regarding the restricted deconstruction of LDPE in the absence of iC$_5$.

Subsequently, we employed this approach with LDPE and iC$_5$ under otherwise identical conditions to follow the transformations (Fig. 5). Initially, new vibrations are assigned exclusively to iC$_5$, whose main characteristics are the $\nu_{as}$(CH$_3$) and $\nu_s$(CH$_3$) bands at 2960 cm$^{-1}$ and 2873 cm$^{-1}$, respectively (cf. Supplementary Fig. 16). The intensities of these CH$_3$ signals increased with the reaction time on stream (see inset in Fig. 5 and Supplementary Fig. 23). Simultaneously, bands assigned to CH$_2$ stretching vibrations of LDPE (2913 cm$^{-1}$ and 2845 cm$^{-1}$) gradually decreased and shifted to 2902 cm$^{-1}$ and 2838 cm$^{-1}$ (due to the impact of polymer chain strain on vibrational frequencies), respectively[48]. These observations reflect the deconstruction of LDPE into short alkanes via cracking-alkylation with iC$_5$. Evidently, the prominent increase of methylene rocking and skeletal stretching of –(CH$_2$)$_n$– in the regions of 760–900 cm$^{-1}$ and 650–680 cm$^{-1}$ agrees with the model for the depolymerization process[48,49]. The bands of [C$_4$Py] cation and dichloromethane are well preserved, indicating that they are largely not involved in the reaction.

It should be noted that the signals of alkene intermediates are not observed during LDPE cracking (compared to the standard alkene in Supplementary Fig. 24), which matched well with in situ $^1$H NMR spectroscopy. To interrogate the presence of alkenes, an iC$_5$ on-off experiment was performed during the operando IR study (Supplementary Fig. 25). After passing N$_2$ at 40 min to purge iC$_5$, the characteristic bands of conjugated dienes at 1630, 1500, 1482, and 1165 cm$^{-1}$ were observed[50]. This indicates that the removal of iC$_5$ leads to a different reaction pathway, forming acid-soluble conjugated oil (ASO) via oligomerization and hydrogen transfer reactions. We concluded, therefore, that the endothermic and exothermic steps of cracking and alkylation occurred closely coupled, i.e., alkenes formed via cracking rapidly are added to the carbenium ion in the alkylation cycle.

Building on these findings, we proposed a set of key elementary steps (Fig. 6 and Supplementary Fig. 26) initiated by the dissociation of Al$_2$Cl$_7^-$ to active catalytic AlCl$_3$ species (Rxn 1). Then, TBC as an initiator, reacted with AlCl$_3$ to generate initial carbenium ions (Rxn 2). All formed carbenium ion-intermediates should be surrounded and stabilized by the counterpart AlCl$_4^-$ anions. Then, AlCl$_4^-$-coordinated carbenium ions as reactive ion-pair intermediates activate the C–H bonds of hydrocarbons (either LDPE or iC$_5$, Rxn 3 and 4), followed by C–H bond cleavage via hydride transfer to preferentially form carbenium ions in the polymer and iC$_5$. Next, the formed polyolefinic carbenium ion pairs undergo isomerization and cracking via β-scission, which yields short carbenium ions and alkenes (Rxn 5). Simultaneously, the formed alkenes react with iC$_5^+$ via the alkylation step, shifting the equilibrium and catalyzing polyolefin conversion (Rxn 6). The long-chain fragments undergo further cracking, and alkylation cycles to the branched alkylate. The PE carbenium ion can be

$$Al_2Cl_7^{\ominus} \; \rightleftharpoons \; AlCl_4^{\ominus} \; + \; AlCl_3 \qquad \text{(Rxn 1:dissociation )}$$

$$AlCl_3 + TBC \; \rightleftharpoons \; AlCl_3 \cdots Cl \cdots C_4 \; \rightleftharpoons \; C_4^{\oplus} \cdots AlCl_4^{\ominus} \qquad \text{(Rxn 2: Cl abstraction)}$$

$$C_n + C_4^{\oplus} \cdots AlCl_4^{\ominus} \; \rightleftharpoons \; C_n^{\oplus} \cdots AlCl_4^{\ominus} + C_4 \qquad \text{(Rxn 3: H transfer)}$$

$$iC_5 + C_4^{\oplus} \cdots AlCl_4^{\ominus} \; \rightleftharpoons \; iC_5^{\oplus} \cdots AlCl_4^{\ominus} + C_4 \qquad \text{(Rxn 4: H transfer)}$$

$$C_n^{\oplus} \cdots AlCl_4^{\ominus} \; \longrightarrow \; C_x^{=} \; + \; C_{n-x}^{\oplus} \cdots AlCl_4^{\ominus} \qquad \text{(Rxn 5: }\beta\text{-scission)}$$

$$C_x^{=} \; + \; iC_5^{\oplus} \cdots AlCl_4^{\ominus} \; \longrightarrow \; C_{x+5}^{\oplus} \cdots AlCl_4^{\ominus} \qquad \text{(Rxn 6: alkylation)}$$

$$C_{x+5}^{\oplus} \cdots AlCl_4^{\ominus} + iC_5 \; \longrightarrow \; C_{x+5} + iC_5^{\oplus} \cdots AlCl_4^{\ominus} \qquad \text{(Rxn 7: H transfer)}$$

**Fig. 6 | Proposed sequence of key reaction steps in the cracking-alkylation of polyolefin with iC5 over [C4Py]Cl-2AlCl3.** All resulting carbenium ion-intermediates are not bare species but surrounded and stabilized by the counterpart AlCl$_4^-$ to form ion pairs.

terminated by recombining hydride (Rxn 7). The termination reactions are the reverse of the initiation reactions.

Thus, the concentration of carbenium ions governs the overall reaction rate, with these intermediates surrounded and stabilized by AlCl$_4^-$ anions, existing as AlCl$_3$ adducts via reversible transformation. The total concentrations of AlCl$_3$ adducts ($\sum$[AlCl$_3$-adduct]) are equivalent to $\Delta$[AlCl$_4^-$]. Initially, the concentration of carbenium ions corresponds to the TBC loading, which explains the direct correlation between the initial reaction rate, the concentration of TBC loading, and the dissociation of Al$_2$Cl$_7^-$ (Supplementary Fig. 11).

Given that cracking and alkylation occurred concurrently and continuously, the initial rate (r) normalized to the total Al adducts concentration can be expressed as Eq. 3 (see detailed derivations in Supplementary Note 3):

$$r_{Al} = \frac{r}{\sum[AlCl_3 - adduct]} = \frac{k_{alkyl}k_{crack}K_{H1}K_{H2}K_{a1}K_{a2}[C_nH_{2n+2}][iC_5H_{12}]}{k_{oligom}K_{H1}K_{a1}[C_nH_{2n+2}] + k_{alkyl}K_{H2}K_{a2}[iC_5H_{12}]} \tag{3}$$

in which [C$_n$H$_{2n+2}$] and [iC$_5$H$_{12}$] are the concentrations of poly-olefin and iC$_5$, respectively; $K_{a1}$ and $K_{a2}$ denote the equilibrium constants of the association of carbenium ion-pair intermediates with polyolefin and iC$_5$, respectively; $K_{H1}$ and $K_{H2}$ denote the equilibrium constants of subsequent hydride transfer steps; $k_{crack}$ and $k_{oligom}$ are the forward and reverse rate constants of the polyolefin cracking step; $k_{alkyl}$ denote the rate constant of the alkylation step. We hypothesize that the effects of the environment of carbenium ion-pairs are identical for reacting molecules, i.e., the association equilibrium constants ($K_a$) and the hydride transfer equilibrium constants ($K_H$) of LDPE and iC$_5$ are identical: $K_{a1} \approx K_{a2}$ and $K_{H1} \approx K_{H2}$, then we have:

$$\frac{1}{r_{Al}} = \frac{1}{k_{crack}K_aK_H[C_nH_{2n+2}]} + \frac{k_{oligom}}{k_{alkyl}k_{crack}K_aK_H[iC_5H_{12}]} \tag{4}$$

Clearly, the rate of tandem cracking-alkylation depends closely on the combination of the carbenium ion-pairs association equilibrium constants ($K_a$), the hydride transfer equilibrium constants between Al active species and reactants ($K_H$), the cracking and alkylation rate constant ($k_{crack}$ and $k_{alkyl}$), and the concentrations of reactants. We also plotted $1/r_{Al}$ against $1/[C_nH_{2n+2}]$ and $1/[iC_5H_{12}]$ as well as $1/[C_{16}]$, respectively (Supplementary Fig. 27). All experiment data agreed well with Eq. 4. The reciprocal of the $1/r_{Al}$ increases linearly with the reciprocal of the concentrations of [C$_n$H$_{2n+2}$] and [iC$_5$H$_{12}$], respectively.

## Discussion

In summary, two active species evolve in chloroaluminate ionic liquids during low-temperature kinetically coupled cracking and alkylation of polyolefins with iso-paraffins. Initially, the main species of chloroaluminate ions during operation are the monomer being generated in situ through the dissociation of Al$_2$Cl$_7^-$ and its reactive adduct with TBC. The adduct undergoes a reversible transformation into a *tert*-butyl carbenium ion and AlCl$_4^-$ ion pair. Subsequently, hydride transfer from the formed *tert*-butyl carbenium ion leads to carbenium ions in the alkane polymer strands and eventually in iC$_5$ (the alkylating agent). Hydride transfer is rapid and facilitated by an environment of high ionic strength. Specifically, the propagation of carbenium ion intermediates by hydride transfer always involves species that are associated and stabilized by AlCl$_4^-$ anions. Computational study strongly supports these conclusions.

In situ $^1$H NMR spectra showed the presence and fluctuation of olefinic intermediates throughout the reaction, which consistently remained at a low concentration of ~0.2–0.5%. The introduction of additional olefin as a substitute for TBC exhibited a pronounced preference for oligomerization over alkylation, resulting in no discernible enhancement for LDPE conversion. These findings are in accordance with the operando IR study and kinetic analysis showing that cracking and alkylation occur concertedly. Carbenium ions stabilized within the alkane polymer strands isomerize and later cleave by β-scission, yielding shorter carbenium ions and alkenes. The alkenes formed participate in the alkylation cycle at very high rates by adding to carbenium ions generated from iC$_5$.

Taken together, this insight enables the advancement of catalytic transformation of polyolefins and will contribute to the design of a next-generation family of catalysts, facilitating the deconstruction of discarded polyolefins with low energy consumption and a reduced carbon footprint.

## Methods
### Chemicals
All chemicals were purchased from Sigma-Aldrich and directly used without any purification: low-density polyethylene (LDPE, average M$_w$ ~ 4000, average M$_n$ ~ 1700; Product-No.:427772), n-hexadecane (n-C$_{16}$H$_{34}$, anhydrous, ≥99%), isopentane (anhydrous, ≥ 99%), N-butylpyridinium chloride (≥ 98%), anhydrous aluminum(III) chloride (99.99 %), trans-decahydronaphthalene (trans-Decalin, 99%), dichloromethane (DCM, ≥ 99.5%), *tert*-butylchlor-ide (TBC, 99%).

## Catalyst characterization

X-ray absorption spectroscopy (XAS) measurements at Al K-edge (1560 eV) were measured on the PHOENIX beamline at the Swiss Light Source (SLS) of the Paul Scherrer Institute (Villigen, Switzerland). All spectra were measured in fluorescence mode using a one-element energy dispersive silicon drift diode (KeteK) detector. The measurements were performed using a PTFE liquid flow cell at room temperature. The design of the cell was adapted from the liquid titration cell available at the PHOENIX I end-station. The cell was equipped with a liquid inlet and outlet and a 500 nm thick silicon nitride window. Various [C4Py]Cl-xAlCl3 ionic liquids with [C4Py]Cl:AlCl3 ratio between 1:1 and 1:2 were prepared inside a glove box. The ionic liquid with [C4Py]Cl:AlCl3 = 1:1 was diluted in DCM as it was a solid at room temperature. Before measurements, the ionic liquids were carefully removed from the glove box with a glass syringe. A syringe pump was then used to fill the liquid flow cell with the ionic liquid. The cell was flushed with DCM followed by nitrogen between two measurements. Beam damage to the samples was assessed by comparing successive measurements and several spectra were averaged to reduce the signal-to-noise ratio. X-ray absorption near edge structure (XANES) and extended x-ray absorption fine structure (EXAFS) analyses were performed using Athena and Artemis software packages[51]. All spectra were corrected for self-absorption using the "fluo" algorithm as implemented in Athena. For XANES analyses, the spectra were normalized and flattened. For EXAFS analyses, the spectra were normalized, background subtracted, $k^2$–weighted, and Fourier-transformed (FT) in the k-range between $k = 2.5$ Å$^{-1}$ and $k = 8.5$ Å$^{-1}$. EXAFS fitting was performed in k-space, simultaneously on the $k^1$–, $k^2$–, and $k^3$–weighted data in the k-range between $k = 2.7$ Å$^{-1}$ and $k = 8.0$ Å$^{-1}$. The spectra for EXAFS analyses were limited to an upper bound of ~1833 eV ($k \sim 8.6$ Å$^{-1}$) due the occurrence of Si K-edge at ~1839 eV.

In situ Raman spectra were measured using a Renishaw inVia Reflex Raman System equipped with a 785 nm RL532C laser. The spectrometer was combined with a Leica DM2700M optical light microscope from Leica Microsystems for focusing the Raman laser. The measurement conditions were identical to those in our tube reactor system.The measurement conditions were identical to our tube reactor system. LDPE (200 mg), iC5(800 mg), TBC (5 mg) and 2 mmol ionic liquids (2 mmol) diluted in 3 ml DCM were added into the borosilicate glass. The reactor was then heated to 60 °C with magnetic stirring at 1200 rpm. Note that there is quick phase separation without stirring, in which the ionic liquid and DCM phase are on the bottom. Therefore, the laser beam was directly focused on the bottom of the glass reactor by a × 5 objective with a maximum laser power of 1.0 mW to avoid laser-induced sample degradation during the measurement. The spectra were collected every 5 min (Exposure time: 1 s, Laser Power: 50 %, accumulations: 20 times) for a total duration of 180 min.

In situ $^{27}$Al magic-angle spinning nuclear magnetic resonance ($^{27}$Al MAS NMR) spectroscopy measurements were conducted on a Varian-Agilent Inova wide-bore 300 MHz NMR spectrometer with a commercial 7.5 mm Vespel pencil-type MAS probe. The spectrometer operated at $^{27}$Al Larmor frequencies of 78.204 MHz. Sample spinning at ~4 kHz enabled the acquisition of high-resolution MAS NMR spectra. A specialized in situ MAS NMR rotor was utilized to ensure the proper sealing of a mixture at elevated temperature and pressure[52]. The sample cell space volume was ~300 μL. For acquiring the $^{27}$Al signal, a single-pulse sequence was utilized, comprising a 5 μs pulse width (equivalent to a pulse angle of 50°), an acquisition time of 50 ms, and a recycle delay of 0.5 s. All spectra were referenced externally to a 1 M Al(NO3)3 (0 ppm) aqueous solution[53].

The $^{27}$Al spectra are deconvoluted into two resonances centered at 103 and 97 ppm through standard background subtraction and fitting the experimental curve to a combination of Lorentzian and Gaussian lines. The integrated line widths were calculated by summing up the full width at half maximum (FWHM) of individual peaks.

Ex situ $^{27}$Al MAS NMR experiments were conducted using a Varian-Agilent Inova 63 mm wide-bore 850 MHz NMR spectrometer. A commercial 3.2 mm pencil-type MAS probe was employed, enabling the use of ~15 mg of sample for each experiment. For acquiring each $^{27}$Al MAS NMR spectrum, a single-pulse sequence was utilized with a pulse length of 0.4 μs, corresponding to a pulse angle of 45°. The experiments had a recycle time of 1 s and a total accumulation of 5000 scans. All spectra were acquired at a sample spinning rate of 20 kHz ± 2 Hz and were referenced to 1.5 M Al(NO3)3 in H2O (0 ppm), using the center of the octahedral peak of solid γ-Al2O3 (at 13.8 ppm) as a secondary reference. To ensure the accuracy of the measurements, the weights of the samples loaded into the MAS rotor were recorded, and four spectra were acquired to verify the stability of the spectrometer. The matching and tuning conditions of the radiofrequency (RF) circuit of the NMR probe were established using a network analyzer. All other experimental conditions were kept identical for all analyzed samples. By normalizing the absolute peak areas to the sample mass and plotting them against the known Al concentration from elemental analysis, a calibration factor was obtained. The spectra were analyzed using the NUTS NMR data processing software (Acorn NMR Inc.).

In situ $^1$H NMR experiments were conducted using a Bruker NEO console and stack, coupled with a 500 MHz Oxford magnet and equipped with a Bruker PI HR-BBO500S2-BBF/H/D-5.0-Z SP probe. The conversion of n-hexadecane (n-C16) was monitored using $^1$H NMR spectroscopy in mixtures of [C4Py]−2AlCl3, iC5, TBC, and CD2Cl2. In a typical experiment, an NMR tube was loaded with [C4Py]−2AlCl3 (100 mg, 0.23 mmol), C16 (35 mg, 0.16 mmol or 2.5 mmol C), and TBC (0.5 μmol, corresponding to 0.2 mol% relative to [C4Py]−2AlCl3). Varying amounts of isopentane (iC5) ranging from 0 to 0.9 mmol were incrementally added under the established reaction conditions while maintaining all other quantities constant. The data were recorded with a 400 MHz spectrometer.

The operando IR spectra were recorded using a ReactIR™ 45 m spectrometer (Mettler Toledo) connected to a 150 mL Parr autoclave at 60 °C. The bottom of the autoclave contains a probing window that is connected to the sentinel probe through the conduit K4, which allows the collection of the operando IR spectra in the liquid phase at different temperatures. Before these tests, a background spectrum was collected at 60 °C under a vacuum. Then, the autoclave reactor was loaded with the LDPE (2.0 g), iC5 (8.0 g), ionic liquids (20 mmol) diluted in30 ml DCM, and TBC (50 mg) as an additive, followed by flushing with nitrogen to remove atmospheric carbon dioxide. The reactor was heated, and the stirring started once the temperature reached 60 °C with a mechanical stirring speed of 700 rpm. The spectra with 4 cm$^{-1}$ resolution were collected every 15 s for a total duration of 240 min, and each spectrum was scanned 256 times.

## Preparation of N-butyl pyridinium chloride-aluminum chloride ionic liquid ([C4Py]Cl-xAlCl3) preparation

A series of [C4Py]Cl-xAlCl3 ionic liquids prepared by mixing N-butyl pyridinium chloride([C4Py]Cl) and anhydrous aluminum chloride at different molar ratios (1.0, 1.2, 1.4, 1.6, 1.8, 2.0 and 2.5) in a glove box. In a typical synthesis of [C4Py]Cl-2AlCl3 ionic liquid, anhydrous aluminum chloride (6 mmol) was slowly added to [C4Py]Cl (3 mmol) in a DURAN® borosilicate glass tube at room temperature, yielding a light-gray liquid[21].

## Catalytic tandem cracking-alkylation reaction of polyolefin with isopentane

The tandem cracking-alkylation reaction of polyolefin with isopentane was carried out in a 30 mL DURAN® borosilicate glass tube (equipped with an open-top screw cap with silicone liner). In a typical reaction, LDPE (200–800 mg) iC5 (200–800 mg), and TBC (5 mg, as an additive) are added into the glass tube containing [C4Py]Cl-xAlCl3 ionic liquids diluted in dichloromethane. Then, the reactor was heated up to

specified temperatures under magnetic stirring (1200 rpm). For the kinetic studies, identical reactions were run in parallel at different time intervals. After the reaction, the glass tube was cooled to −25 °C, resulting in a biphasic system. Notably, a scaled-up reaction to 150 ml Parr autoclave produced identical results, allowing operando IR measurements (See Supplementary Fig. 21).

The headspace of the reactor was first analyzed using a gas chromatograph equipped with Supel-Q™ PLOT fused silica capillary column (30 m × 0.53 mm × 2.0 μm) and a mass spectrometer. Then, an additional 2 ml of isopentane as an extracting agent was injected into the tube (cooled to −25 °C), enabling a complete phase separation. The ionic liquid phase under the bottom was then transferred to a glass vial and quenched by NaOH aqueous solution at low temperature, in which organic residues were extracted by chloroform and analyzed by GC-mass spectrometry. The organic phase was then mixed with 15 mg of trans-decalin as an external standard and 1 ml chloroform for promoting the mixing of organic products, followed by GC and GC-MS analyses. The unreacted solids were separated by filtration to quantify the solid conversion. The sum of all detected products was taken as the total yield. The response factor for alkanes of each carbon number and the corresponding retention time were calibrated using n-alkane standard ($C_7$- $C_{40}$, Supeco-49452-U). The unreacted LDPE suspended solids were separated from the top phase by a simple filtration to quantify the conversion.

Converson, selectivity, yield and the initial rate (r) were calculated accdording to the following equations:

$$Conversion = \left(1 - \frac{mass\ of\ residual\ LDPE}{initial\ mass\ of\ LDPE}\right) \times 100\% \quad (5)$$

$$Yield(C_i,\%) = \left(\frac{\sum Area(C_i)}{Area(external\ standard)} \times \frac{mass\ of\ external\ standard}{initial\ mass\ of\ LDPE}\right) \times 100\% \quad (6)$$

$$r = \frac{initial\ mass\ of\ LDPE - mass\ of\ residual\ LDPE}{reaction\ time(t)} \quad (7)$$

where $\sum Area(C_i)$ is the sum of GC areas of $C_i$, in which all peaks between n-$C_{i-1}$ and n-$C_i$ were assumed as branched $C_i$ alkanes unless GC-MS suggested otherwise. Area (external standard) is the GC area of the external standard. Note that all products and external standard are saturated hydrocarbons; the relative response factors for each species were assumed to be 1.0. We studied the rates at low conversions <20 wt.%, and the carbon balances were >95 %.

The concentration of $AlCl_3$ in [$C_4Py$]Cl-$xAlCl_3$ ($x \in [1,2]$) at spectific reaction time (t) was calculated using Eqs. 1–2 in the main text (see detailed derivations in Supplementary Note 2).

### Density functional theory calculations for XAS and Raman
All density functional theory (DFT) calculations were carried out using ORCA quantum chemistry package version 5.3[54]. Geometry optimizations and vibrational frequency calculations utilized Ahlrichs def2-TZVP basis sets and auxiliary basis sets SARC/J[55,56]. The B3LYP hybrid-GGA functional was employed for all calculations, along with the atom pairwise dispersion correction and the Becke-Johnson damping scheme (D3BJ)[57]. To enhance computational efficiency, the RIJCOSX approximation was also applied[58]. X-ray absorption spectra of the optimized structures were simulated using time-dependent density functional theory (TDDFT) with the Tamm-Dancoff approximation, allowing for transitions from Al 1 s orbitals. Up to 150 roots were calculated, allowing for transitions from Al 1 s orbitals. The calculated intensities include electric dipole, magnetic dipole and electric quadrupole contributions. All calculations were performed with a conductor-like polarizable continuum model (CPCM) using DCM as solvent[58].

### Density functional theory calculations for $^{27}$Al MAS NMR
Quantum chemistry calculations of $^{27}$Al chemical shifts were utilized to aid in the interpretation of NMR spectra using the Amsterdam Density Functional (ADF) software[59,60]. The cluster models were optimized using the Beck-Lee-Yang-Parr functional[61], and a Slater-type, all electron, triple- ζ, two-polarization function (TZ2P) was utilized[62]. To correlate the calculated nuclear shielding from DFT to the observed NMR chemical shifts, [$C_4Py$]Cl-1$AlCl_3$ was used as the computational standard at 103 ppm, serving as a secondary reference to 1.0 M $Al(NO_3)_3$ (0 ppm) due to its unique narrow peak identified in experiments. The calculated nuclear shielding of other models was converted to an experimental chemical shift scale using the formula $\delta_{obs}$ = 443.84- $\delta_{calc}$ + 103.0 ppm.

### AIMD simulations and reaction free energy calculations
To achieve a well-equilibrated structure at 70 °C, a model system was constructed, comprising 1 n-$C_{16}H_{34}$, 2 [$C_4Py$]Cl-2$AlCl_3$ ionic liquids ([$C_4Py$]$^+$-$Al_2Cl_7^-$), 6 i$C_5$, 1 TBC, and 105 DCM in a 23 Å cubic cell, mimicking experimental compositions. Subsequently, DFT-based ab initio molecular dynamics (AIMD) simulations were conducted with periodic boundary conditions (3D PBC) using the generalized gradient approximation (GGA) with Perdew, Burke, and Ernzerhoff (PBE) exchange-correlation functionals[63], and Grimme's third-generation (DFT-D3) dispersion correction[64], as implemented in the CP2K package[65]. The core electrons were described using Goedecker−Teter−Hutter (GTH) pseudopotentials[66], while the valence wavefunctions were expanded in optimized double-ζ Gaussian basis sets. An auxiliary plane wave basis set with a cutoff energy of 400 Ry was employed for the calculation of the electrostatic terms. Due to the substantial size of the supercell, the Γ-point approximation was used for the Brillouin zone integration.

AIMD simulations were performed by sampling the NVT canonical ensemble with a Nosé−Hoover thermostat chain, using a time step of 0.5 fs at 70 °C. After ~60 ps of simulation, an equilibrated structure was obtained from the last frame. Subsequently, AIMD-based Blue Moon ensemble calculations[67] were performed by constraining the distance between two specified atoms from the reacting molecules. The integration of the ensemble-averaged force due to the constraints along the reaction coordinate generated a free energy profile as a function of distance, providing the standard Helmholtz free energy (ΔF°) and activation energy ($E_a$).

## Data availability
All data are available within the article and its Supplementary Information files and from the authors upon request. Source data are provided in this paper.

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

## Acknowledgements

We thank Dr. Camelia N. Borca from the Phoenix beamline at the Swiss Light Source (SLS) of the Paul Scherrer Institute in Switzerland for their assistance in Al XAS characterization. We also thank G. L. Haller (Yale University), J. G. Chen (Columbia University), S. L. Scott (UC Santa Barbara), and M. L. Sarazen (Princeton University) for their discussion of the manuscript and helpful suggestions. J.A.L., W.Z., S.K., L. H., W.H., J.M., B.Y., O.Y.G., D.R., J.F., D.M.C., J.H., H.W., and M.L. thank the U.S. Department of Energy (DOE), Office of Science, Office of Basic Energy Sciences (BES), Division of Chemical Sciences, Geosciences and Biosciences (towards a polyolefin-based refinery: understanding and controlling the critical reaction steps, FWP 78459) for funding support. J.H also thank the U.S. Department of Energy (DOE), Office of Science, Office of Basic Energy Sciences (BES), Division of Chemical Sciences, Geosciences and Biosciences (Multifunctional Catalysis to Synthesize and Utilize Energy Carrier, FWP 47319) for funding support of in situ NMR work. Computational work was performed using the National Energy Research Scientific Computing Center located at the Lawrence Berkley National Laboratory provided by a user proposal and the Research Computing Facility at PNNL.

## Author contributions

W.Z. and J.A.L. conceived the research; W.Z., C.Y., Y.S., and P.Z. performed the experiments and the in-situ Raman and operando IR characterizations. L.H., J.H., S.K., and W.H. provided data on the in-situ NMR measurements; R.K. and L.W. performed the XAS characterization; M.-S.L. and D.R. performed AIMD simulations along with the Blue Moon ensemble method. J.F., H.W., O.Y.G., D.M.C., J.M., and B.Y. cooperated with the discussion and provided valuable suggestions. The manuscript was written through the contributions of all authors. All authors have approved the final version of the manuscript.

## Competing interests

The Authors declare no competing interests.
