## [Peer Review File · Nature Communications]

Active species in chloroaluminate ionic liquids catalyzing low-temperature polyolefin deconstructionREVIEWER COMMENTS

Reviewer #1 (Remarks to the Author):

The article is based on numerous measurement results obtained by different methods. The composition of chloroaluminate species and their role in the required low-temperature activation and subsequent tandem cracking alkylation for polyolefin conversion in the presence of isoparaffins will be elucidated. The mechanism is a matter of speculation, leaving many aspects unconsidered.

In order to clarify the mechanism, the Al species were observed during the reaction. An Al₃ species was calculated from the missing difference of the detected AlCl₄⁻ and Al₂Cl₇⁻. Evidence is said to have been provided by DFT calculations. The influence of adducts TBC on the initial reaction rate was observed. The degree of conversion of the polymer was determined by filtering off the unreacted polymers. By varying the i-butane and the polymer, a reaction order was established with respect to these components. A hyperbolic rate approach was formulated.

The state of the art is insufficiently addressed. With regard to cleavage and polymerisation with AlCl₃ as an active component in homogeneous and heterogeneous catalysis, there is an extensive state of the art that should be urgently addressed. Only by discussing these results it is possible to make a self-critical judgement of the present measurement results. Even review papers are available.

It is surprising that the Al species are consumed or formed over time during the reaction. There is thus no stationary equilibrium between the Al species over the course of time. Theoretically, a catalyst or the active component should not be consumed. At the start of the reaction, where the reaction rate is at its highest, the active component is not yet present, but only Al₂Cl₇⁻. This raises the question of the extent to which time effects lead to a misinterpretation of the results.

The Al-adduct, which is referred to as active, increases sharply from the start of the reaction.

Determining the kinetics under these conditions is therefore highly error-prone and must be investigated more intensively in order to make reliable statements. Reaction orders of 0.35 or 0.45 and the hyperbolic rate approach must be strongly questioned, even if the fit suggests this.

The formed olefins could also interact with AlCl₃ and the depolymerization can also occur via reaction with this species as AlCl₄⁻ was found to be inert. TBC may only act as an initiator. More extensive measurements at different conditions (olefin content, temperatures) should be carried out. The changing concentration of the aluminium species must be clarified, which could be due to interactions of formed olefins. The statements made are otherwise purely speculative and do not provide any additional knowledge.

However, alternative possibilities with regard to the reaction process are certainly conceivable. Olefins could influence the equilibrium of the Al species. The extent to which the solvent, the products formed (especially olefins) and i-butane have an effect on the complex is unclear. It should be shown more clearly when and under what conditions which statements were made (what was measured and under what conditions? And what conclusion is drawn from this).

It was claimed that alkylation should be very fast because no olefins can be observed in the product spectrum. If olefins interact with AlCl₃, then this observation would not be surprising. Unfortunately, the product spectrum is not shown in the article, neither as a sum nor over time, although this could provide valuable information about the reaction process over time. A mass balance could be used to draw conclusions about the extent to which the polymer or oligomers formed are dissolved in the reaction mixture. The reference to DFT results does not help here because it is not clear what the reaction mixture consists of. As it is assumed that the olefins formed interact with the catalyst until they are alkylated, experiments should be carried out with the additional addition of olefins at the start of the reaction.

The decomposition of LDPE and alkylation of the decomposition products was investigated. The extent to which such a process can be used for recycling is more than questionable because the authors ignore the difference between plastic/plastic waste (which contains impurities with heteroatoms and toxins) and pure polymer. The sentence "Containing solely carbon and hydrogen, they would be a

clean feed, minimising the need for costly heteroatom removal" is therefore misleading. The extent to which the technology presented is suitable for recycling real plastic waste must therefore be questioned until it has been proven on a real plastic waste mixture.

In der Conclusion taucht der Satz "Excellent hydride transfer in the cascade cracking-alkylation of LDPE and iC5 leads to a relatively narrow product distribution and minimizes the formation of red oil waste." Auf, der irritiert, weil keine Ergebnisse diesbezüglich gezeigt wurden. Grundsätzlich geht der Reviewer davon aus, dass eine geschlossene Argumentation zur Klärung der Forschungsfrage vorgenommen wird und belastbare Argumente klar erklärt werden, sodass mit dem Artikel der Wissensgewinn gut dokumentiert wird.

Ebenso irritiert aus besagten Gründen „Overall, the high activity of the hydrocarbons involved is caused by high density of ions (high ionic strength). The outstanding hydride transfer ability in this environment maintains the narrow product distribution. Together this insight allows to advance the catalytic transformation of the polyolefins and will help to design of a next-generation family of catalysts enabling robust upcycling of discarded polyolefins with low energy consumption and carbon footprint.“

The question arises as to what catalytic conditions mean in this context. "The Raman spectrum of [C4Py]Cl-xAlCl₃ in the presence of TBC (5 mol% TBC relative to the [C4Py]Cl-xAlCl₃ loading, which is three times higher than under catalytic conditions)"

The question also arises as to what "Notably, the small C-H stretching signals of terminal CH₃ supports that LDPE cracking is equilibrium-limited (Supplementary Fig. 16c), which agrees well with the in situ ²⁷Al NMR results" means and whether this relatively small change in the signal is not caused by the reactant (penetration depth).

Fig. 1 is unclear because alkylation is mentioned in different places. A step-by-step formulation of the reaction mechanism, which should be the aim of the work, would be more helpful. This would make the nebulous formulations more precise.

Overall, the article has many ambiguities and a rigorous proof excluding other possible explanations through specific modified experiments is missing.

Reviewer #2 (Remarks to the Author):

This manuscript explores the mechanism involved in the conversion of low-density polyolefin plastic using chloroaluminate ionic liquids as a catalyst, building upon previous work by the same research group. Employing a wide array of experimental techniques such as ²⁷Al NMR, Raman, IR, and X-ray absorption spectroscopy, alongside computational methods, DFT-NMR and AIMD, the study delves into the active species during the LDPE reaction.

Initially challenging to grasp, the manuscript necessitated multiple readings. The experimental data does not support the dissociation of Al₂Cl₇⁻ into AlCl₃ and AlCl₄⁻, a phenomenon only backed by simulations. The absence of concrete experimental evidence from various spectroscopic techniques raises doubts about the involvement of AlCl₃.

The proposed adduct (AlCl₃...Cl...iC₄) is backed by AIMD simulations, yet its presence remains undetected in spectroscopic evidence. Furthermore, the lack of its structural depiction in the reaction scheme implies incomplete understanding, particularly regarding its interaction with AlCl₃.

The conversion of LDPE, with or without the addition of TBC, still progresses, indicating the formation of the carbenium ion. However, the possibility of an alternate mechanism in the absence of TBC

remains unexplored. The question arises whether the mechanism remains consistent when ionic liquids are introduced to LDPE without TBC.

While the paper utilises numerous spectroscopic techniques to speculate on the mechanism, the absence of experimental data confirming the presence of the adduct or AlCl_3 leads to debate regarding their involvement in the mechanism.

The manuscript can be accepted based on the quality of both experimental and simulation data, however, the following changes must be implemented prior to its publication:

1. Improve the clarity of the manuscript, clearly and logically explain information gained from each technique, and how they relate to full understanding.
2. This is a good work, and it does not need overinterpretation to be valuable. Rewrite the manuscript to clearly state that simulations suggest the mechanism proposed, but the evidence from experimental methods could not be obtained. It is absolutely ok to leave it at that. Science is a journey in understanding and it is much more valuable to honestly outline the state of knowledge gained from this large quantity of research, than overinterpret this.
3. **Clearly and accurately** explain what has been learned from experiments with and without TBC, and provide critical analysis what remains unknown/unproven for each mechanism. Provide this in the conclusion part of the paper. This will fuel further research and inspire good quality papers to follow.

Response to reviewers' comments

We sincerely value the insightful comments provided by the reviewers and the editor. In response, we have made revisions to address the majority of the suggestions and have emphasized these changes within the manuscript. Below, we present a point-by-point response to the reviewers' comments and concerns.

Reviewer #1 (Remarks to the Author):

[**Comments**] The article is based on numerous measurement results obtained by different methods. The composition of chloroaluminate species and their role in the required low-temperature activation and subsequent tandem cracking alkylation for polyolefin conversion in the presence of isoparaffins will be elucidated. The mechanism is a matter of speculation, leaving many aspects unconsidered. In order to clarify the mechanism, the Al species were observed during the reaction. An AlCl_3 species was calculated from the missing difference of the detected AlCl_4^- and Al_2Cl_7^- . Evidence is said to have been provided by DFT calculations. The influence of adducts TBC on the initial reaction rate was observed. The degree of conversion of the polymer was determined by filtering off the unreacted polymers. By varying the i-butane and the polymer, a reaction order was established with respect to these components. A hyperbolic rate approach was formulated.

Response:

Thank you for your time and effort in reviewing the manuscript and providing constructive suggestions. We make clarifications in this revision and performed supplementary experiments for new evidence. Please see below our point-to-point responses to your comments.

[**Comments**] The state of the art is insufficiently addressed. With regard to cleavage and polymerisation with AlCl_3 as an active component in homogeneous and heterogeneous catalysis,

there is an extensive state of the art that should be urgently addressed. Only by discussing these results it is possible to make a self-critical judgement of the present measurement results. Even review papers are available.

It is surprising that the Al species are consumed or formed over time during the reaction. There is thus no stationary equilibrium between the Al species over the course of time. *Theoretically, a catalyst or the active component should not be consumed.* At the start of the reaction, where the reaction rate is at its highest, the active component is not yet present, but only Al_2Cl_7^- . This raises the question of the extent to which time effects lead to a misinterpretation of the results.

The Al-adduct, which is referred to as active, increases sharply from the start of the reaction. Determining the kinetics under these conditions is therefore highly error-prone and must be investigated more intensively in order to make reliable statements. Reaction orders of 0.35 or 0.45 and the hyperbolic rate approach must be strongly questioned, even if the fit suggests this.

Response:

Despite significant research and industrial application of chloroaluminate-based ionic liquids, especially in the alkylation of isoparaffins with olefins, the identification of catalytically active species and the quantification of active sites within chloroaluminate ions—exhibiting a variety of nuclearities during operation—remain elusive (*Chem. Soc. Rev.*, 2008, 37, 123; *Ind. Eng. Chem. Res.* 2020, 59, 15811). Moreover, the challenge arises from the dual action of the chloroaluminate ions and alkyl chloride additive that initiate the reaction to maintain a specific concentration of carbenium ions and simultaneously facilitate hydride transfer. The present paper discusses several key points regarding the speciation of the active catalyst component. It aims to provide a clearer mechanistic insight and highlight its implications for the overarching catalytic chemistry.

i) Variations in chloroaluminate anions indicate that Al_2Cl_7^- acts not as the active species

We fully agree with the reviewer that a catalyst must not be consumed within a catalytic cycle. The reviewer is reminded, however, that chemistry occurring in parallel to the catalytic cycle can increase or decrease the concentration of the catalytically active species.

We previously prepared a series of $[\text{C}_4\text{Py}]\text{Cl}-x\text{AlCl}_3$ ionic liquids with the varying molar ratio of AlCl_3 and $[\text{C}_4\text{Py}]\text{Cl}$. Increasing the initial fraction of anhydrous AlCl_3 promotes the conversion from AlCl_4^- to Al_2Cl_7^- , resulting in higher reaction rates. However, the initial reaction rates of $[\text{C}_4\text{Py}]\text{Cl}-x\text{AlCl}_3$ does not correlate directly with the concentration of Al_2Cl_7^- (**Fig. R1**). Instead, the rate of catalyzed reaction was proportional to the initial ratio of $[\text{Al}_2\text{Cl}_7^-]$ to $[\text{AlCl}_4^-]$. Therefore, we hypothesized that the increasing reactivity is proportional to an AlCl_3 species, which is generated through the in situ dissociation of the dimeric Al_2Cl_7^- ($\text{Al}_2\text{Cl}_7^- \rightleftharpoons \text{AlCl}_4^- + \text{AlCl}_3$). To test this hypothesis, we employ in situ ^{27}Al MAS NMR and in situ Raman spectroscopy to assess the dynamic evolution of chloroaluminate species in ionic liquids in the present manuscript.

Fig. R1. Relationship between initial LDPE conversion rate, the mole fraction (χ) of Al_2Cl_7^- , and empirical $[\text{AlCl}_3]$ concentration ($K * [\text{AlCl}_3]$). This figure is added as Supplementary Fig .1 in the revised supporting information.

ii) How initiator (i.e., TBC) affects the speciation of chloroaluminate anions during reaction

Alkyl chloride additives are essential in chloroaluminate-based ionic liquid-catalyzed alkylation of isoparaffins with olefins. Numerous studies have attributed the effectiveness of these additives to the in situ generation of HCl (ISOALKY™ Technology, *Ind. Eng. Chem. Res.* 2020, 59, 15811), functioning similarly to classical Brønsted acid-catalyzed alkylation processes, such as those involving hydrofluoric acid (HF) or sulfuric acid (H₂SO₄). Aschauer et al. showed that halide additives act as carbenium ion initiator that directly generates the carbenium ions for the alkylation reaction. In contrast, with Brønsted-acids this species is indirectly formed through a hydride transfer between protonated butene and isobutane (*Catal. Lett.*, 2011, 141, 1405-1419). Our earlier research indicated that adding protic co-reagents (e.g., H₂O) did not alter reaction rates, ruling out that HCl is directly acting as catalytically active species. In contrast, the addition of tert-butyl chloride (TBC) significantly enhanced LDPE conversion. This observation agreed well with the report of Aschauer et al., demonstrating that TBC acts as a carbenium ion initiator.

We used in situ ²⁷Al MAS NMR and in-situ Raman spectroscopy to obtain a comprehensive assessment of the dynamic evolution of chloroaluminate species in ionic liquids in the present manuscript. While TBC acts as a source of initial carbenium ions for reaction initiation, the mechanism by which TBC facilitates carbenium ion formation, and its interaction with chloroaluminate ions (as ion-pairs or adducts) still needs clarification.

Dynamic evolution of chloroaluminate species evidenced by in situ/operando techniques during the reaction.

In this study, we monitored the dynamic transformation of chloroaluminate species within ionic liquids using in situ Raman spectroscopy throughout the reaction. Initially, the chloroaluminate species in [C₄Py]Cl-2AlCl₃ ionic liquids predominantly existed as dimeric Al₂Cl₇⁻ anions

([C₄Py]Cl + AlCl₃ → C₄Py⁺⋯Al₂Cl₇⁻). Considering the charge and Al balance (C₄Py⁺⋯Al₂Cl₇⁻ → C₄Py⁺⋯AlCl₄⁻ + AlCl₃), variations in the concentrations of Al₂Cl₇⁻ and AlCl₄⁻ can be reasonably used to quantitatively assess the concentration of formed AlCl₃. The overall concentration of [AlCl₃] at specific reaction time can be expressed as: $\sum[\text{AlCl}_3]_t = \Delta[\text{Al}_2\text{Cl}_7^-]_t = \Delta[\text{AlCl}_4^-]_t$.

Clearly, the variation of chloroaluminate species showed a close correlation with LDPE conversion (**Fig. 3b**): the Al₂Cl₇⁻ species was rapidly consumed, and its fraction decreased by approximately 0.3 when 40% of LDPE was converted. The remaining LDPE (60%) only consumed ~0.1 of Al₂Cl₇⁻. This was also observed with the variation of Al₂Cl₇⁻ and AlCl₄⁻ using *n*-C₁₆H₃₄ as a reactant instead of LDPE (**Supplementary Fig. 14**). We concluded that the reaction is initiated by TBC reacting with AlCl₃ that is in situ generated from the dissociation of Al₂Cl₇⁻.

The resulting Al species predominantly exist as organic/inorganic adduct, as otherwise the presence of AlCl₄⁻ (together with carbenium ions) would be reflected by a distinct Raman signal specific to AlCl₄⁻. This observation is concomitant with the rapid generation of AlCl₃ species originating from the dissociation of Al₂Cl₇⁻ during the initial stages of the reaction (i.e., shifting the equilibrium through the adduct formation), eventually reaching the equilibrium.

In response to the reviewer's comments, we have expanded and clarified the discussion in the revised manuscript, incorporating the suggested points. We also conducted a systematic investigation of the [C₄Py]Cl-xAlCl₃ ionic liquids (x=1.2, 1.4, 1.6, 1.8, 2.0) with the varying molar ratio of AlCl₃ and [C₄Py]Cl, for catalyzing the tandem cracking-alkylation of LDPE-iC₅ (**Fig. R2**). This study included different amounts of TBC additive (1, 2, 5, 10, 20, 30 mg) to explore their influence on the reaction dynamics. The concentration of AlCl₃-adducts, formulated as

$[\text{AlCl}_3 \cdots \text{Cl} \cdots \text{C}_4] = \frac{[\text{Al}_2\text{Cl}_7^-] \cdot [\text{TBC}]}{[\text{AlCl}_4^-] \cdot K}$, was examined under these conditions. Our analysis, plotted on a

logarithmic scale of $\text{Log}(r)$ against $\text{Log}\left(\frac{[\text{Al}_2\text{Cl}_7^-][\text{TBC}]}{[\text{AlCl}_4^-]}\right)$, showed a relationship that closely follows first-order kinetics. Additionally, it was observed that the initial reaction rate is directly proportional to the concentration of TBC loading, indicating a significant dependence on the concentration of this additive.

We also agree with the referee that certain kinetic data, such as turnover frequency and reaction orders, may lead to confusion due to the dynamic variation in active species with varying TBC additive. Therefore, we have decided to omit this part to avoid any possible misunderstandings by readers.

Fig. R2. a. Illustration of chloroaluminate transformation in the presence of TBC. b. The initial LDPE conversion rate as a function of $\frac{[\text{Al}_2\text{Cl}_7^-][\text{TBC}]}{[\text{AlCl}_4^-]}$ concentration where the concentration was determined by varying the molar ratio of initial anhydrous AlCl_3 and $[\text{C}_4\text{Py}]\text{Cl}$, and TBC amount. This figure is added as Supplementary Fig. 11 in the revised supporting information.

[Comments] The formed olefins could also interact with AlCl_3 and the depolymerization can also occur via reaction with this species as AlCl_4^- was found to be inert. TBC may only act as an initiator.

More extensive measurements at different conditions (olefin content, temperatures) should be carried out. The changing concentration of the aluminium species must be clarified, which could be due to interactions of formed olefins. The statements made are otherwise purely speculative and do not provide any additional knowledge.

However, alternative possibilities with regard to the reaction process are certainly conceivable. Olefins could influence the equilibrium of the Al species. The extent to which the solvent, the products formed (especially olefins) and i-butane have an effect on the complex is unclear. It should be shown more clearly when and under what conditions which statements were made (what was measured and under what conditions? And what conclusion is drawn from this).

It was claimed that alkylation should be very fast because no olefins can be observed in the product spectrum. If olefins interact with AlCl_3 , then this observation would not be surprising. Unfortunately, the product spectrum is not shown in the article, neither as a sum nor over time, although this could provide valuable information about the reaction process over time. A mass balance could be used to draw conclusions about the extent to which the polymer or oligomers formed are dissolved in the reaction mixture. The reference to DFT results does not help here because it is not clear what the reaction mixture consists of. As it is assumed that the olefins formed interact with the catalyst until they are alkylated, experiments should be carried out with the additional addition of olefins at the start of the reaction.

Response:

We thank the reviewer for the constructive comments. We make clarifications in this revision and performed supplementary experiments for new evidence. In specifics to the two comments:

1) The addition of olefins does not influence the equilibrium of aluminium species, nor does it act as a promoter for the tandem cracking-alkylation reaction.

We first studied the interaction between olefin and [C₄Py]Cl-2AlCl₃. The spectra of [C₄Py]Cl-2AlCl₃ remained virtually unchanged in the presence of varying concentrations of pentene (C₅⁼) as well as in its absence (**Fig. R3**). This observation implies that olefin incorporation does not promote the dissociation of Al₂Cl₇⁻. Subsequently, to further probe the influence of olefins on the tandem cracking-alkylation of LDPE and iC₅, C₅⁼ was introduced as a substitute for TBC additive. As shown in **Fig. R4**, the introduction of C₅⁼ gave only 18 wt.% conversion to LDPE exhibiting no discernible enhancement when compared to the baseline conditions that lacked such additives. In contrast, the addition of TBC significantly increased the initial reaction rate, resulting in 100% of LDPE conversion within 180 mins. Our AIMD-based Blue Moon ensemble simulation confirmed this by showing a highly endothermic process ($\Delta F^\circ=125$ kJ/mol) responsible for forming the AlCl₄⁻ and AlCl₃-C₅⁼ adduct through interaction between Al₂Cl₇⁻ and C₅⁼ (**Fig. R5**).

We found that the addition of C₅⁼ notably favors both the alkylation reaction with iC₅ and oligomerization as well as disproportionation (**Fig. R6a**), which in agree well with the findings reported by Aschauer *et al.* for C₄ alkylation (**Fig. R6b**, from *Catal. Lett.*, 2011, 1405-1419). AlCl₃-alkene adduct can be formed through the interaction between the alkene's π -bond electrons and the empty p-orbital of AlCl₃ (*ChemistryOpen* 2020, 9, 662-666), which can then undergo further competitive hydride transfer (alkylation) and oligomerization. The reaction initiates when a carbenium ion is initially formed either through chloride abstraction from TBC or through the protonation of C₅⁼ by a Brønsted acid (**Fig. R6c**). Notably, the alkylation occurs at a considerably slower rate compared to the oligomerization of alkenes. Consequently, alkylation in industrial practice typically need to maintain significant excesses of isoparaffins over alkenes, with ratios ranging approximately from 7 to 10 (not considering the back-mixed reactor at high olefin conversion shifting tis ratio well above 100), to mitigate the competitive oligomerization of olefins

into acid-soluble oils (ASO). These by-products not only emerge undesirably but also contribute to catalyst deactivation. Additionally, when olefins are maintained at a high concentration, it becomes essential to sustain a high concentration of Brønsted acid within the alkylation framework. This is beneficial for the protonation of alkenes, which effectively stabilizes carbenium ions, facilitating their reintegration into the propagation cycle.

Fig. R3. **a.** Raman spectra of $[\text{C}_4\text{PyCl}]\text{-}2\text{AlCl}_3$ ionic liquids in the presence of pentene ($\text{C}_5^=$). and **b,** The corresponding variation of chloroaluminate species. Note: spectra recorded at room temperature with varying $\text{C}_5^=$ amount (0-0.2mmol) to 2 mmol $[\text{C}_4\text{PyCl}]\text{-}2\text{AlCl}_3$ ionic liquids. **This figure is added as Supplementary Fig .18 in the revised supporting information.**

Fig. R4. a. Time-resolved conversion profile of LDPE in the presence of TBC or C_5^- as additives, compared to conditions without additives. Reaction conditions: LDPE 200 mg, iC_5 800 mg, $[C_4Py]Cl-AlCl_3$ 3 mmol, additive 0.05 mmol, DCM 3 mL, 70 °C. This figure is added as Supplementary Fig .19 in the revised supporting information.

Fig. R5. Computed reaction pathway for the formation of $AlCl_3-C_5^-$ adduct. Plotted is the standard Helmholtz free energy (ΔF°) as a function of the internuclear distance of Al-H for the reaction of $Al_2Cl_7^-$ and C_5^- to form $AlCl_4^-$ and a $AlCl_3-C_5^-$ adduct. ΔF° is calculated using the Blue Moon ensemble approach for ab initio molecular dynamics. Included are representative structures along the reaction pathway. This figure is added as Supplementary Fig .20 in the revised supporting information.

Fig. R6. **a.** Alkylation of $C_5^=$ with iC_5 over $[C_4Py]Cl-2AlCl_3$ in the presence of H_2O or TBC as additives. Reaction conditions: $C_5^=$ 100 mg, isopentane (iC_5) 800 mg, $[C_4Py]Cl-2AlCl_3$ 1 mmol, additive 0.05 mmol, DCM 3 mL, 25 °C. **b.** Alkylation of $C_4^=$ with iC_4 over $[BMIM]Cl-AlCl_3$ in the presence of tert-butyl halides (Cl/Br) and H_2O as additive, compared with sulfuric acid-catalyzed alkylation. Data from published literature from Aschauer et al. *Catal Lett*, 2011,141:1405–1419. **c.** The proposed reaction pathways for the reaction of $C_5^=$ with iC_5 involve several key steps that reflect the complexities of hydrocarbon transformations.

2) The olefins produced were not distinctly detected by gas chromatography (GC) and infrared (IR) spectroscopy, not because of interaction with $AlCl_3$, but rather due to their low concentration.

In the tandem cracking-alkylation of LDPE and iC_5 , alkenes are hardly observed in the product stream, only traces of propene (< 0.1 wt. %) were found in the headspace. Actually, alkenes formed

in the cracking cycle via β -scission. Afterwards, a rapid alkylation of alkene/isoparaffin significantly shifts the equilibrium via exothermic alkylation of alkene fragments. It should be noted that in the industrially practiced alkylation (2-butene and iso-butane reacting) alkenes are hardly observed in the product stream.

We then employed in situ ^1H NMR spectroscopy to monitor the alkene formation during the tandem-alkylation and cracking of n-hexadecane (C_{16}) and $i\text{C}_5$ in the ionic liquid medium. For a typical experiment, an NMR tube was charged with $[\text{C}_4\text{PyCl}]\text{-2AlCl}_3$ (0.23 mmol), C_{16} (0.16 mmol or 2.5 mmol C), and TBC (0.5 μmol). Varying amounts of $i\text{C}_5$ were added under the standard reaction conditions previously described to investigate the impact of the alkylating agent on alkene detection. **Fig. R7a** showed the variation in the concentration of unsaturated hydrogen atoms within the olefinic region (4.4–6.5 ppm). This variability indicates the presence and fluctuation of olefinic intermediates throughout the reaction. The ratio of unsaturated chains remains constant, maintaining a low concentration approximately between 0.2–0.5% throughout the reaction (**Fig. R7b**).

Additionally, IR analysis reveals that the characteristic signals of alkenes are absent in the presence of $i\text{C}_5$. However, in the absence of $i\text{C}_5$, distinct bands corresponding to conjugated dienes (**Fig. R8**) at 1630 cm^{-1} , 1500 cm^{-1} , 1482 cm^{-1} , and 1165 cm^{-1} became apparent. These findings are consistent with those reported by Li et al. in their study titled "Characterization and hydrogenation removal of acid-soluble oil in ionic liquid catalysts for isobutane alkylation (*Ind. Eng. Chem. Res.* 2021, 60, 38, 13764–13773)".

Thus, we deduce that alkenes, generated synchronously during the cracking cycle, engage in very rapid alkylation reactions with $i\text{C}_5$ (addition to the carbenium ion). This interaction significantly

alters the equilibrium via the exothermic alkylation of alkene fragments, thus, effectively suppressing the oligomerization of olefins.

Fig. R7. In situ ^1H NMR spectra of tandem cracking-alkylation of $n\text{-C}_{16}$ and $i\text{C}_5$ over $[\text{C}_4\text{Py}]\text{Cl}-2\text{AlCl}_3$. **a**, The representative in situ ^1H NMR spectra varying $n\text{-C}_{16}$ and $i\text{C}_5$ loading. **b**, the fraction of total alkenes correlates with time. Conditions: Hexadecane ($n\text{-C}_{16}$) 0.16 mmol, isopentane ($i\text{C}_5$) 0-0.90 mmol, $[\text{C}_4\text{Py}]\text{Cl}-2\text{AlCl}_3$ 0.23 mmol, TBC 0.5 μmol , CD_2Cl_2 , 25 $^\circ\text{C}$, 4 h. This figure is added as Supplementary Fig. 17 in the revised supporting information.

Fig. R8. Operando IR spectra of the *iC*₅ on-off experiment. Conditions were as follows: LDPE, 2 g; *iC*₅, 8 g; [C₄Py]Cl-2AlCl₃, 20 mmol; and temperature, 60 °C. Data were acquired with an automatic sample scan interval of 15s over the range of 4000-700 cm⁻¹. This figure is added as Supplementary Fig .24 in the revised supporting information.

[Comments] The decomposition of LDPE and alkylation of the decomposition products was investigated. The extent to which such a process can be used for recycling is more than questionable because the authors ignore the difference between plastic/plastic waste (which contains impurities with heteroatoms and toxins) and pure polymer. The sentence "Containing solely carbon and hydrogen, they would be a clean feed, minimizing the need for costly heteroatom removal" is therefore misleading. The extent to which the technology presented is suitable for recycling real plastic waste must therefore be questioned until it has been proven on a real plastic waste mixture.

Response:

We agree with the reviewer. In the revised version of our manuscript, we have corrected the sentence to avoid any misunderstanding. Polyolefin, ideally consisting solely of carbon and hydrogen, often requires the addition of various impurities and additives in real-world applications to enhance its properties, performance, stability, and appearance. Our current study is focused on

the low-temperature catalytic transformations conducted in a single-stage process, deliberately sidelining the complexities introduced by feedstock purity to primarily focus on proof-of-concept demonstrations. While fillers and additives might influence the catalytic activity, the underlying catalytic chemistry remains consistent across different types of polymers when utilizing such catalysts. The primary distinction among various polymers revolves around the differential accessibility for the formation of carbenium ions. It should be noted in passing that we have shown that “real polymer waste” though sorted is converted in identical manner (Science, 2023, 379,807-811)

[Comments] In der Conclusion taucht der Satz “Excellent hydride transfer in the cascade cracking-alkylation of LDPE and iC5 leads to a relatively narrow product distribution and minimizes the formation of red oil waste.” Auf, der irritiert, weil keine Ergebnisse diesbezüglich gezeigt wurden. Grundsätzlich geht der Reviewer davon aus, dass eine geschlossene Argumentation zur Klärung der Forschungsfrage vorgenommen wird und belastbare Argumente klar erklärt werden, sodass mit dem Artikel der Wissensgewinn gut dokumentiert wird.

Ebenso irritiert aus besagten Gründen „Overall, the high activity of the hydrocarbons involved is caused by high density of ions (high ionic strength). The outstanding hydride transfer ability in this environment maintains the narrow product distribution. Together this insight allows to advance the catalytic transformation of the polyolefins and will help to design of a next-generation family of catalysts enabling robust upcycling of discarded polyolefins with low energy consumption and carbon footprint.

Response: We thank the reviewer for highlighted these sentences. We have eliminated them to prevent to any misunderstanding. While we have these data inclusion would exceed the volume and scope of the current manuscript.

[Comments] The question arises as to what catalytic conditions mean in this context. "The Raman spectrum of $[\text{C}_4\text{Py}]\text{Cl-xAlCl}_3$ in the presence of TBC (5 mol% TBC relative to the $[\text{C}_4\text{Py}]\text{Cl-xAlCl}_3$ loading, which is three times higher than under catalytic conditions)"

Response:

We would like to clarify that the conventional concentration of the TBC additive utilized in the reaction conditions was approximately 1.6 mol%, relative to the $[\text{C}_4\text{Py}]\text{Cl-xAlCl}_3$ loading. Subsequent Raman spectroscopic analysis has demonstrated that increasing the TBC concentration to 5% within ionic liquids does not alter the spectral profiles (Fig. R9). This observation suggests that the presence of TBC does not influence this equilibrium between Al_2Cl_7^- and AlCl_4^- , as any perturbation would be expected to manifest through a distinct Raman signature characteristic of AlCl_4^- . We have reformulated the sentences in the revised version.

Fig. R9. Raman spectroscopy analysis of the interaction between TBC and [C₄Py]Cl-xAlCl₃. a, Raman spectra of neat [C₄Py]Cl-xAlCl₃ ionic liquids. b. Raman spectra of [C₄Py]Cl-xAlCl₃ ionic liquids in the presence of TBC. Spectra recorded at room temperature with TBC to [C₄Py]Cl-xAlCl₃ molar ratio of 5 %, . c. Plot of the Raman area ratio of Al₂Cl₇⁻ versus the molar ratio of AlCl₃/[C₄Py]Cl (x). This figure is added as Supplementary Fig. 12 in the revised supporting information.

[Comments] The question also arises as to what "Notably, the small C-H stretching signals of terminal CH₃ supports that LDPE cracking is equilibrium-limited (Supplementary Fig. 16c), which agrees well with the in situ ²⁷Al NMR results" means and whether this relatively small change in the signal is not caused by the reactant (penetration depth).

Response:

Supplementary Figure 21c (previously Figure 16c in the earlier supporting information) presents the operando IR spectra obtained during the deconstruction of LDPE in the absence of iC₅, using [C₄Py]Cl-2AlCl₃ as the catalyst at 60 °C. The appearance of weak signals corresponding to terminal CH₃ groups indicates the limited LDPE deconstruction due to the endothermic cleavage of the C–C bonds, which is thermodynamically unfavorable at lower temperatures. This observation is corroborated by in situ ²⁷Al NMR spectra recorded for LDPE deconstruction, reinforcing the conclusion regarding the limited deconstruction of LDPE in the absence of iC₅.

[Comments] Fig. 1 is unclear because alkylation is mentioned in different places. A step-by-step formulation of the reaction mechanism, which should be the aim of the work, would be more helpful. This would make the nebulous formulations more precise.

Response:

We agree with the reviewer that Scheme 2 (not Fig.1) is challenging for readers. To mitigate this, we added the sequence of key reaction steps in the cracking-alkylation of polyolefin with *i*C₅ over [C₄Py]Cl-2AlCl₃ (Fig. R10). The reaction is initiated by the dissociation of Al₂Cl₇⁻ to active catalytic AlCl₃ species (Rxn 1). Then, TBC as an initiator reacted with AlCl₃ to generate initial carbenium ions (Rxn 2). All formed carbenium ion-intermediates will be surrounded and stabilized by the counterpart AlCl₄⁻ anions. Then, AlCl₄⁻-coordinated carbenium ions as reactive ion-pair intermediates activate the C–H bonds of hydrocarbons (either LDPE or *i*C₅, Rxn 3 and 4), followed by C–H bond cleavage via hydride transfer to preferentially form carbenium ions in the polymer and *i*C₅. Next, the formed polyolefinic carbenium ion pairs isomerize and crack via β-scission, which yields short carbenium ions and alkenes (Rxn 5). Simultaneously, the formed alkenes reacted with *i*C₅⁺ via the alkylation step, shifting the equilibrium and catalyzing polyolefin conversion (Rxn 6). The long-chain fragments undergo further cracking, and alkylation cycles to the branched alkylate. The carbenium ion can be terminated by hydride transfer (Rxn 7). The termination reactions are the reverse of the initiation reactions.

Fig. R10. Proposed sequence of key reaction steps in the cracking-alkylation of polyolefin with *i*C₅ over [C₄Py]Cl-2AlCl₃. This encompasses the proposed reaction pathways for the cracking of a polyolefin coupled

with the alkylation of the resulting olefin intermediate, including initiation of carbenium ions, the progression of cracking and alkylation cycles, as well as the termination of carbenium ions. This figure is added as Scheme 2 in the revised main text.

[**Comments**] Overall, the article has many ambiguities and a rigorous proof excluding other possible explanations through specific modified experiments is missing.

Response:

We hope that this revised manuscript with clarification on the novelty and significant readership, improved logical flow, and enhanced discussion that is founded in new evidence suffice for publication in Nature Communications.

Reviewer #2 (Remarks to the Author):

This manuscript explores the mechanism involved in the conversion of low-density polyolefin plastic using chloroaluminate ionic liquids as a catalyst, building upon previous work by the same research group. Employing a wide array of experimental techniques such as ^{27}Al NMR, Raman, IR, and X-ray absorption spectroscopy, alongside computational methods, DFT-NMR and AIMD, the study delves into the active species during the LDPE reaction. Initially challenging to grasp, the manuscript necessitated multiple readings. The experimental data does not support the dissociation of Al_2Cl_7^- into AlCl_3 and AlCl_4^- , a phenomenon only backed by simulations. The absence of concrete experimental evidence from various spectroscopic techniques raises doubts about the involvement of AlCl_3 .

The proposed adduct ($\text{AlCl}_3 \cdots \text{Cl} \cdots \text{iC}_4$) is backed by AIMD simulations, yet its presence remains undetected in spectroscopic evidence. Furthermore, the lack of its structural depiction in the

reaction scheme implies incomplete understanding, particularly regarding its interaction with AlCl_3 .

The conversion of LDPE, with or without the addition of TBC, still progresses, indicating the formation of the carbenium ion. However, the possibility of an alternate mechanism in the absence of TBC remains unexplored. The question arises whether the mechanism remains consistent when ionic liquids are introduced to LDPE without TBC.

While the paper utilizes numerous spectroscopic techniques to speculate on the mechanism, the absence of experimental data confirming the presence of the adduct or AlCl_3 leads to debate regarding their involvement in the mechanism.

The manuscript can be accepted based on the quality of both experimental and simulation data, however, the following changes must be implemented prior to its publication:

1. Improve the clarity of the manuscript, clearly and logically explain information gained from each technique, and how they relate to full understanding.
2. This is a good work, and it does not need overinterpretation to be valuable. Rewrite the manuscript to clearly state that simulations suggest the mechanism proposed, but the evidence from experimental methods could not be obtained. It is absolutely ok to leave it at that. Science is a journey in understanding and it is much more valuable to honestly outline the state of knowledge gained from this large quantity of research, than overinterpret this.

Response (1) and (2): We greatly appreciate the comments from the reviewer regarding our work. We followed, therefore, the suggestion of the reviewer and rewrote the paper with a clearer picture and the consequence for the overall catalytic chemistry.

- 1) We have streamlined the description of the pure ionic liquids' characterization and emphasized the quantitative relationship between the formation of monomer and dimer aluminate species in ionic liquids. This revision aims to provide a clearer understanding of the ionic liquid structure and its relevance to the observed catalytic behaviors.
 - i) To streamline the main text and improve overall comprehensibility, the X-ray absorption spectroscopy analysis, previously presented in Figure 1 of the original submission, has now been relocated to the supporting information as new Supplementary Figure 1.
 - ii) The original statement, "Supplementary Fig. 7 shows the energy barriers for interconversion between Al_2Cl_7^- and AlCl_4^- ($\text{Al}_2\text{Cl}_7^- + \text{AlCl}_4^- \rightarrow \text{AlCl}_4^- + \text{Al}_2\text{Cl}_7^-$) are significantly low (~ 14 kJ/mol), with a reaction free energy of 9 kJ/mol. This interconversion is remarkably more favorable compared to Al_2Cl_7^- dissociation, which exhibits a higher activation energy of 59 kJ/mol", has been removed to ensure clarity in the presentation of our findings.
- 2) We rewrote the part of results and discussion. We first use in situ ^{27}Al NMR and Raman spectroscopy to understand the interaction between the additives (e.g., TBC and olefin) and $[\text{C}_4\text{Py}]\text{Cl}-2\text{AlCl}_3$, further upholding by AIMD simulations (Fig. 2b and Supplementary Fig. 20). The findings indicate that the incorporation of TBC and olefins does not markedly alter the dissociation behavior of Al_2Cl_7^- , as compared to conditions where they are absent. This suggests that the interaction between Al_2Cl_7^- with TBC results in limited conversion, indicative of a reversible catalytic process. Furthermore, we hypothesize that additives, such as TBC, form more stable organic adducts with AlCl_3 .
Next, we studied the dynamic transformation of these chloroaluminate species within ionic liquids, focusing on their role in catalyzing the tandem cracking-alkylation of LDPE (n-C₁₆)

with iC_5 , with/without TBC additive. Specifically, we observed that the $Al_2Cl_7^-$ species experienced rapid consumption in the presence of TBC during the initial stages of the reaction. This phenomenon suggests the generation of $AlCl_3$ species originating from the dissociation of $Al_2Cl_7^-$ ($Al_2Cl_7^- \rightarrow AlCl_4^- + AlCl_3$), which subsequently reached equilibrium as the reaction progressed.

3. **Clearly and accurately** explain what has been learned from experiments with and without TBC, and provide critical analysis what remains unknown/unproven for each mechanism. Provide this in the conclusion part of the paper. This will fuel further research and inspire good quality papers to follow.

Response:

Following the reviewer's suggestion, we have incorporated discussions on the mechanism both with and without TBC, as well as olefin additives, aiming to enhance our understanding of the proposed mechanism.

Carbenium ion chain mechanism unaltered: regardless of additive presence

The addition of TBC as the carbenium ion initiator markedly influences the reaction rate but does not alter the reaction mechanism. In chloroaluminate ionic liquids catalyzed system, the dominating $Al_2Cl_7^-$ reacts with TBC via chloride abstraction, resulting in the formation of the $AlCl_3$ adducts ($AlCl_3 \cdots Cl \cdots iC_4$). These adducts rapidly convert into reactive ion-pair species, i.e., *tert*-butyl carbenium ions and $AlCl_4^-$. The initial carbenium ions subsequently activate the C-H bonds of reacting molecules (LDPE and iC_5) through successive hydride transfers, promoting the propagation of carbenium ions.

It is known that DCM reacts with Lewis acids, i.e., is polarized by anhydrous AlCl_3 , forming an electron donor-acceptor complex ($\text{AlCl}_3 \leftarrow \text{ClCH}_2\text{Cl}$). (*Chem. Rev.* 2007, 107, 2037) This complex can further convert into a chloromethyl-carbenium ion and AlCl_4^- pair (*J. Org. Chem.*, 1990, 55, 1224), represented as $[\text{CH}_2\text{Cl}]^+[\text{AlCl}_4]^-$ that initiates the reaction following carbenium ion chemistry (*Catal. Lett.* 2011, 141, 1405). Building on these findings (while our manuscript was under review) we directly utilized anhydrous AlCl_3 as a catalyst (*Angew. Chem. Int. Ed.* 2024, e202319580), demonstrating higher activity compared to corresponding ionic liquids without requiring the addition of TBC as the carbenium ion initiator (**Fig. R11a**). Computational simulations showed that Al_2Cl_7^- react with TBC ($\Delta G^\circ = 23 \text{ kJ/mol}$), exhibiting an equilibrium constant of 10^{-3} (**Fig. R11b**), which is significantly higher than its reaction with the DCM solvent which shows an equilibrium constant of 10^{-5} ($\Delta G^\circ = 31 \text{ kJ/mol}$, **Fig. R11c**). The high concentration of AlCl_3 (give a comparison to the calculated concentration in the ionic liquid) (over)compensates the low-rate constant of DCM compared to TBC and induces higher rates than found with the ionic liquids.

In situ ^{27}Al MAS NMR spectra were employed to identify AlCl_3 -adducts present in DCM (**Fig. R11c**). Signals observed at 0 ppm were attributed to solid AlCl_3 powder. Upon the addition of DCM, two distinct peaks were observed at 92 ppm and 99 ppm, corresponding to Al_2Cl_6 species and the AlCl_3 -DCM adduct, respectively. An excess of solid AlCl_3 in DCM led to prominent peaks at 0 ppm and 99 ppm, indicating increased formation of the AlCl_3 -DCM complex ($\text{AlCl}_3 \leftarrow \text{ClCH}_2\text{Cl}$) via the equilibrium: Al_2Cl_6 (92 ppm) + $2\text{CH}_2\text{Cl}_2 \rightleftharpoons 2\text{AlCl}_3\text{-CH}_2\text{Cl}_2$ (99 ppm). This observation aligns with previous findings regarding associations in AlCl_3 -arene solutions (*Ind. Eng. Chem. Res.*, 2021, 60, 1155). Therefore, we concluded that the absence of detected

adducts in chloroaluminate ionic liquids is attributed to interference from the multinuclearity of chloroaluminate ions.

Fig. R11. (a) Comparison of initial reaction rate over anhydrous AlCl_3 , and $[\text{C}_4\text{Py}]\text{Cl}-2\text{AlCl}_3$ ionic liquid (with/without TBC additive). Reaction conditions: 200 mg LDPE, 800 mg $i\text{C}_5$, 3 ml DCM. Catalyst: 0.5 mmol AlCl_3 vs. 2 mmol $[\text{C}_4\text{Py}]\text{Cl}-2\text{AlCl}_3$ (with TBC 0.05 mmol). (b-c) Ab initio molecular dynamics simulations (Blue Moon ensemble method) of Al_2Cl_7^- reacting with tert-butyl chloride (TBC) and dichloromethane (DCM) to form AlCl_3 complexes, respectively. Images illustrate atomic configurations of the initial and final states. (d) In situ ^{27}Al MAS NMR on anhydrous AlCl_3 in DCM solutions at room temperature.

REVIEWERS' COMMENTS

Reviewer #1 (Remarks to the Author):

The team of authors has put a lot of effort into the revision. The focus is much clearer. The article is now written in a less speculative way. Additional information support the statements made. It now concentrates on the role of the active species and leaves room for further investigations where aspects have not yet been fully clarified. The risk of over-interpreting the results has been significantly reduced.

A concrete reference and categorization of the own results to the state of the art with regard to AlCl_3 species in catalysis in general would still be recommended

In principle, statements about up-cycling should be made with greater caution, because polyamides and water in real plastic waste can massively influence the activity of such catalysts. The mention of deactivation by such substances is a must for a real application, as otherwise a false impression could arise.

The sentence: "Containing solely carbon and hydrogen, they would be a clean feed, minimizing the need for costly heteroatom removal." In the introduction is still misleading.

Also, in the conclusion the sentence: "Taken together, this insight enables the advancement of catalytic transformation of polyolefins and will contribute to the design of a next-generation family of catalysts, facilitating robust upcycling of discarded polyolefins with low energy consumption and a reduced carbon footprint." does not do justice to the fact that plastic waste is also contaminated with heteroatoms from colors, additives and composites, despite being purely polyolefinic plastics.

Reviewer #2 (Remarks to the Author):

This manuscript represents a continuation of prior research conducted by the same research group, focusing on unravelling the mechanism underlying the conversion of low-density polyolefin plastic employing chloroaluminate ionic liquids as catalysts. It is evident that the authors possess a comprehensive grasp of the interplay between experimental and simulated data without overstatement. Additionally, the manuscript adeptly addresses the absence of detectable monomeric AlCl_3 analytically due to its presence as an organic adduct, demonstrates a nuanced understanding of the ionic liquid's structure and the coordination environment surrounding the aluminium centre, as corroborated by ^{27}Al NMR and Raman spectroscopy conducted at various temperatures.

Moreover, the manuscript effectively explains the role of tert-butyl chloride (TBC) and the interaction between AlCl_3 and TBC, supported by in situ ^{27}Al NMR analysis during the catalytic deconstruction of LDPE using $[\text{C}_4\text{Py}]\text{Cl}-2\text{AlCl}_3$, and provides clearer insights into energy barriers from simulations and the initial reaction rate when TBC is used. The mechanism and kinetics governing the catalytic deconstruction of LDPE are explained, with the observation of less than 0.1% alkene serving as compelling evidence for concurrent reactions, further supported by the additional reactions involving isopentene. The proposed reaction sequence is coherently articulated, with each step underpinned by evidence presented earlier in the manuscript.

Overall, this manuscript represents a commendable advancement, embodying significant revisions that culminate in a robust scientific contribution to the field. Therefore, the manuscript can be accepted for publication.

Response to reviewers' comments

Reviewer #1 (Remarks to the Author):

The team of authors has put a lot of effort into the revision. The focus is much clearer. The article is now written in a less speculative way. Additional information support the statements made. It now concentrates on the role of the active species and leaves room for further investigations where aspects have not yet been fully clarified. The risk of over-interpreting the results has been significantly reduced. A concrete reference and categorization of the own results to the state of the art with regard to AlCl_3 species in catalysis in general would still be recommended.

Response: We thank the reviewer for recommending publication of our manuscript in Nature Communications. Based on your recommendations, we have added several references and included the following sentences in the introduction on Page 3, relating to the state of the art concerning AlCl_3 species in catalysis.

“Additionally, alkyl chloride additives are essential in chloroaluminate-based ionic liquid-catalyzed alkylation of isoparaffins with olefins. Numerous studies have attributed the effectiveness of these additives to the in-situ generation of HCl ,²⁶ functioning similarly to classical Brønsted acid-catalyzed alkylation processes, such as those involving hydrofluoric acid (HF) or sulfuric acid (H_2SO_4).²⁷ While Jess and coworker showed that halide additives as the carbenium ion initiator that directly generate the carbenium ions for the alkylation reaction, in contrast to Brønsted-acid, where this species is indirectly formed through a hydride shift between protonated butene and isobutane.²⁸ The challenge lies in elucidating the dual roles of chloroaluminate ions, especially speculative monomeric AlCl_3 , and the alkyl chloride additive, in the carbenium ion-based mechanism that drives the interconnected cracking and alkylation cycles.”

Reference:

- 26 Timken, H. K., Luo, H., Chang, B.-K., Carter, E. & Cole, M. in *Commercial Applications of Ionic Liquids* (ed Mark B. Shiflett) 33-47, doi: 10.1007/978-3-030-35245-5_2 (2020)
- 27 Feller, A. & Lercher, J. A. in *Adv. Catal.* 48, 229-295 (Academic Press, 2004).
- 28 Aschauer, S., Schilder, L., Korth, W., Fritschi, S. & Jess, A. Liquid-Phase Isobutane/Butene-Alkylation Using Promoted Lewis-Acidic IL-Catalysts. *Catal. Lett.* **141**, 1405, doi:10.1007/s10562-011-0675-2 (2011).

In principle, statements about up-cycling should be made with greater caution, because polyamides and water in real plastic waste can massively influence the activity of such catalysts. The mention of deactivation by such substances is a must for a real application, as otherwise a false impression could arise. The sentence: “Containing solely carbon and hydrogen, they would be a clean feed, minimizing the need for costly heteroatom removal.” In the introduction is still misleading. Also, in the conclusion the sentence: “Taken together, this insight enables the advancement of catalytic transformation of polyolefins and will contribute to the design of a next-generation family of catalysts, facilitating robust upcycling of discarded polyolefins with low energy consumption and a reduced carbon footprint.” does not do justice to the fact that plastic waste is also contaminated with heteroatoms from colors, additives and composites, despite being purely polyolefinic plastics.

Response: We appreciate these comments and would like to make a few clarifications below.

- 1) we have eliminated the term 'up-cycling' and instead use 'deconstruction' to prevent any misunderstanding.
- 2) To avoid confusion, we removed the misleading sentence from the introduction: 'Containing solely carbon and hydrogen, they would be a clean feed, minimizing the need for costly heteroatom removal.'

3) We modified the sentence in the conclusion to read: 'Taken together, this insight enables the advancement of catalytic transformation of polyolefins and will contribute to the design of a next-generation family of catalysts, facilitating the **deconstruction** of discarded polyolefins with low energy consumption and a reduced carbon footprint.'

Reviewer #2 (Remarks to the Author):

This manuscript represents a continuation of prior research conducted by the same research group, focusing on unravelling the mechanism underlying the conversion of low-density polyolefin plastic employing chloroaluminate ionic liquids as catalysts. It is evident that the authors possess a comprehensive grasp of the interplay between experimental and simulated data without overstatement. Additionally, the manuscript adeptly addresses the absence of detectable monomeric AlCl_3 analytically due to its presence as an organic adduct, demonstrates a nuanced understanding of the ionic liquid's structure and the coordination environment surrounding the aluminium centre, as corroborated by ^{27}Al NMR and Raman spectroscopy conducted at various temperatures.

Moreover, the manuscript effectively explains the role of tert-butyl chloride (TBC) and the interaction between AlCl_3 and TBC, supported by in situ ^{27}Al NMR analysis during the catalytic deconstruction of LDPE using $[\text{C}_4\text{Py}]\text{Cl}-2\text{AlCl}_3$, and provides clearer insights into energy barriers from simulations and the initial reaction rate when TBC is used. The mechanism and kinetics governing the catalytic deconstruction of LDPE are explained, with the observation of less than 0.1% alkene serving as compelling evidence for concurrent reactions, further supported by the additional reactions involving isopentene. The proposed reaction sequence is coherently articulated, with each step underpinned by evidence presented earlier in the manuscript. Overall, this manuscript represents a commendable advancement, embodying significant revisions

that culminate in a robust scientific contribution to the field. Therefore, the manuscript can be accepted for publication.

Response: We thank the reviewer for the positive remarks.